# Microelectromechanical Transducer to Monitor High-Doses of Nuclear Irradiation

**DOI:** 10.3390/s21175912

**Published:** 2021-09-02

**Authors:** Julien Philippe, Muriel Ferry, Samuel Charlot, Sandrine Assié, Aurélie Lecestre, Guillaume Libaude, André Ferrand, Patrick Pons, Hervé Aubert

**Affiliations:** 1French National Centre for Scientific Research (CNRS), Laboratory for Analysis and Architecture of Systems (LAAS), University of Toulouse, Institut National Polytechnique de Toulouse (INPT), 7 Avenue du Colonel Roche, 31031 Toulouse, France; scharlot@laas.fr (S.C.); sassie@laas.fr (S.A.); aurelie.lecestre@laas.fr (A.L.); glibaude@laas.fr (G.L.); ppons@laas.fr (P.P.); aubert@laas.fr (H.A.); 2Université Paris-Saclay, CEA, Service d’Etude du Comportement des Radionucléides, 91191 Gif-sur-Yvette, France; muriel.ferry@cea.fr; 3Institut Clément Ader (ICA), Université de Toulouse, Modeling of Systems and Mechanical Microsystems (MS2M), 31400 Toulouse, France; andre.ferrand@insa-toulouse.fr

**Keywords:** passive sensors, microelectromechanical systems (MEMS) technology, microelectromechanical transducer, dosimeter, nuclear irradiation, nuclear dose, high-density polyethylene, gas emission yield factor G

## Abstract

This paper reports the design, fabrication and measured performance of a passive microelectromechanical transducer for the wireless monitoring of high irradiation doses in nuclear environments. The sensing device is composed of a polymer material (high-density polyethylene) sealed inside a cavity. Subjected to ionizing radiation, this material releases various gases, which increases the pressure inside the cavity and deflects a dielectric membrane. From the measurement of the deflection, the variation of the applied pressure can be estimated, and, in turn, the dose may be determined. The microelectromechanical structure can also be used to study and validate the radiolysis properties of the polymer through its gas emission yield factor. Measurement of the dielectric membrane deflection is performed here to validate on the one hand the required airtightness of the cavity exposed to doses about 4 MGy and on the other hand, the functionality of the fabricated dosimeter for doses up to 80 kGy. The selection of appropriate materials for the microelectromechanical device is discussed, and the outgassing properties of the selected high-density polyethylene are analysed. Moreover, the technological fabrication process of the transducer is detailed.

## 1. Introduction

Passive (or “zero-power”) and wireless dosimeters show promising features for monitoring irradiation doses in nuclear environments. Indeed, as they do not require battery or/and energy harvesters for supplying DC power, such sensors are expected to have very long lifetimes. Moreover, their wireless interrogation may be performed to remotely estimate the irradiation doses in inaccessible areas. Zero-power and wireless dosimeters may be very useful in irradiated regions that drastically limit any in-situ human being interventions, e.g., the structure health monitoring of nuclear infrastructures after a nuclear incident. Hydrogen Pressure Dosimeters (HPD) [1,2] or track etched detectors [3], as well as sensors using alanine-polymer film [4] or based on radio-photo-thermo-luminescent structures [3,5,6], have been reported so far. However, these devices do not provide real-time measurement of irradiation doses and provide only post-factum monitoring of doses. Moreover, complex methods are often applied during the measurement process, such as electron spin resonance spectroscopy for alanine-based dosimeters [4]. Passive HPD dosimeters using microelectromechanical systems (MEMS) technology and a polymer as reactive material under irradiation have been recently proposed in [7] using a long-range wireless radio-frequency sensing technique that exceeds 3.5 m [8]. These dosimeters may be advantageously used for the remote and real-time measurement of irradiation doses. Very preliminary results are reported in [7] on these very promising devices, but to date proof-of-concept is not established.

This paper investigates for the first time the radiolysis properties of polyethylene by using the MEMS prototype reported in [9] when irradiation doses reach about 80 kGy. The design, constitutive materials and manufacturing process of this MEMS device are discussed, and the integrity of the microelectromechanical transducer is tested for doses up to 4 MGy. A method is also proposed to estimate the irradiation dose and finally, the measured performances of the fabricated MEMS device are compared with theoretical predictions.

## 2. General Working Principle of Passive Radiofrequency Dosimeter

Investigation of the impact of nuclear irradiations on polymers is crucial to guarantee the materials’ compatibility with specific nuclear environments or facilities [10,11,12,13,14,15], to secure nuclear waste packages during transportation, storage and/or disposal [16,17,18,19,20], and finally to improve the physical and chemical properties of the materials [21,22,23,24,25]. The effects of nuclear irradiation on polymeric structures are well-documented [21,22,26,27,28,29] and mainly consist of chains scissions and crosslinking, creation of unsaturated bonds in polymers and generation of low molecular weight components which can be trapped in the materials or released as gases. Their relative contribution depends on the polymer type and irradiation conditions (e.g., the dose level, atmosphere properties, irradiation type, among others) [12,16,28,30,31,32,33]. Following [7,8] the overpressure measurement resulting from polymer outgassing can be used to estimate the irradiation dose. As sketched in Figure 1a, the passive Radio Frequency (RF) dosimeter based on overpressure measurement is composed of a polymer material, which is sealed inside an airtight cavity and releases gases from radiolysis when subjected to ionizing radiation. The outgassing generates an overpressure in the cavity that deflects a thin dielectric membrane electromagnetically coupled with a two-port RF planar resonator below the dielectric membrane. Such deflection modifies the electromagnetic coupling, leading to variation of the scattering parameters of the RF resonator and finally modifies the radar cross-section of the dosimeter (see [34,35,36]). By monitoring of the radar cross-section variation, the measurement of dose can be remotely derived. Aperture-coupled microstrip lines can be advantageously used to feed the membrane-loaded resonator by electromagnetic waves [37,38].

We focus here on the design, fabrication and measured performances of the passive microelectromechanical transducer (which does not include the RF resonator) shown in Figure 1c for the wireless monitoring of high doses of γ-rays up to 80 kGy in nuclear environments.

## 3. Material Choice for the Microelectromechanical Transducer

### 3.1. Constitutive Materials

As illustrated in Figure 1a, the transducer uses two substrates that are bonded together to seal the polymer and RF resonator in the same airtight structure. The bonding steps in the technological process are performed carefully in order to avoid any alteration of the outgassing property of used materials under irradiation. Various assembling techniques can be applied to seal the polymer and RF resonator in the same airtight structure [39], such as eutectic bonding [40], silicon [41,42] or metal-to-metal [43,44] direct bonding, solder bonding [45] or low-temperature melting glass bonding [46]. The anodic bonding [47] consists of the direct silicon-to-glass bonding performed at high temperature (typically between 300–600 °C) and high voltages (between 0.2 kV and 2 kV) between contacting substrates. During the technological process, the bonding interface is fabricated from strong siloxane Si-O-Si bonds. Details of the bonding mechanisms and chemical reactions can be found in [47,48]. This process is widely applied to package components and devices [49,50], but also to fabricate sealed cavities [48,51,52,53] with long-term airtightness [8,54] and low leak rates [52]. The airtightness quality depends mainly on the surface state of the substrates, the temperature and applied voltages. Studies reported in the open scientific literature have demonstrated that a good resistance to γ irradiations (γ-rays) of substrates can be obtained from isolated or assembled silicon and borosilicate substrates using anodic bonding [55]. No significant change in the physical properties was reported on the silicon substrate [55], while a slight densification and presence of optical alterations were detected on the borosilicate material [56,57,58,59]. However, to the best knowledge of the authors, no study has been reported on the impact of irradiation on the bonding interface and, more specifically, on the airtightness of sealed cavities. Since borosilicate glass is mainly composed of Si-O bonds, γ-rays are expected to weakly impact the bonding interface. Furthermore, since anodic bonding was previously used for the fabrication of sensors working under ionizing radiation in spatial environment [49,60,61] or placed in irradiated areas [7,8,53], this bonding technique is a priori suitable for the fabrication of passive dosimeters. Consequently, high resistivity silicon (HR Si) [62] and borofloat B33 substrates [63] are chosen for the membrane and resonator substrates, respectively. The low losses of these substrates make them convenient for sensing applications in the microwave frequency range [8,34,38,64].

### 3.2. Polymer

As mentioned before, outgassing is one phenomenon among many other reactions when polymers are subjected to ionizing radiations. Both the nature and quantity of gases depends on the polymer composition and the irradiation properties, such as the irradiation temperature, the surrounding atmosphere, the nature of the irradiation (γ-rays, pile irradiations and ions irradiations, among others), the dose deposition or the dose rate [16,17,19,65]. In our application, the polymer must fulfil the following requirements:(i)It must be suitable for the selected assembling technological process, which requires good thermal stability with no loss of mass, and stable radiolysis properties in the temperature range of the anodic bonding.(ii)It must generate gases over a wide range of irradiation dose levels and in sufficient quantity to be measured.(iii)Polymers, including aromatic cycle or unsaturated C=C bonds, should be discarded because of their higher stability under irradiation, leading to reduced outgassing when compared with other families of polymers [22,30,32].(iv)Emission by the irradiated polymer of corrosive gases must be avoided in order to maintain the integrity of the transducer structure. Undesirable hydrochloric or hydrofluoric acids are actually released during the radiolysis of polyvinyl chloride and polyvinylidene fluoride [16,22,32].

Among the polymers studied in the literature, polyethylene (PE) is a good candidate for our application. PE is one of the most stable polymers as temperature varies: mass loss and polymer degradation start near 250 °C in air and at about 400 °C in an inert atmosphere [66,67,68]. This polymer presents the thermal stability required by the process fabrication process. According to the state-of-the-art, the hydrogen emission yield factor $G_{H_{2}}$ of PE is between 2.3 × 10^−7^ mol·J^−1^ and 4.6 × 10^−7^ mol·J^−1^ [33]. Two main types of PE are commercially available: *high-density polyethylene* (HDPE) and *low-density polyethylene* (LDPE). HDPE is composed of long chains with very few branches compared with LDPE (see Figure 2). The higher the quantity of branches in PE, the lower the crystallinity. Unlike claims reported in some studies (e.g., [30,69]) it appears that the crystallinity degree of polyethylene has a weak impact on the hydrogen emission yield factor [33,70]. Crystallinity was not a discriminant parameter in the conditions of our application. Furthermore, PE can be used up to few MGy [10,69] and generates very high quantities of gases under irradiation [11,12,16,19,28,32]. The gases generated by HDPE during irradiation under vacuum or an inert atmosphere are mainly composed of hydrogen $H_{2}$ (between 85% and 99%). In case of LDPE, a small quantity of hydrocarbon gas (<7%) may also be present, such as ethane $C_{2}H_{6}$ (<2.3%), methane $CH_{4}$ (<0.9%) and/or ethylene $C_{2}H_{4}$ (<0.4%), these gases being related to the branched chains nature and concentration. To facilitate and ascertain the estimation of the quantity of gases released from PE, HDPE appears to be the best choice. When irradiation is performed under an oxidative atmosphere, oxygen reacts with the radical in the polymer chain to form a peroxy radical, which then reacts to form stable defects [71], including carbon monoxide (CO) and dioxide ($CO_{2}$) [11,12,17,30,72,73]. This oxygen to estimate precisely the total pressure inside the cavity. Consequently, the irradiation of the PE polymer sealed in a vacuum appears to be much more preferable. Moreover, there is no dose rate effect under such atmospheric irradiation conditions.

The gas emission yield factor $G_{gas}$ of HDPE in an inert atmosphere is close to the hydrogen emission yield factor $G_{H_{2}}$. This factor is estimated in Section 7.3 for the selected polymer and for different dose levels, dose rates and fixed (room) temperature. HDPE can be found in many different shapes, such as rods, pellets, powders, films, sheets or fibres [74,75]. Even if the hydrogen emission yield factor $G_{H_{2}}$ seems not to depend on the HDPE thickness and shape [76], the release of this gas in the surrounding atmosphere depends on its diffusion inside the material (Fick’s law of diffusion). Thin films of PE appear to be the best choice in our application. As the hydrogen emission yield factor $G_{H_{2}}$ is almost constant for low doses up to 200 kGy [30], the quantity of $H_{2}$ is directly proportional to the dose in this range. At room temperature and in an inert (He) atmosphere, the decrease of factor $G_{H_{2}}$ occurs at higher doses due to the generation of C=C unsaturated bonds during the irradiation that acts as an energy sink [30,70,77]: energy and radicals transfer preferentially to these bonds, leading to undesirable reduction of $H_{2}$ release. For example in [70], the hydrogen emission yield factor $G_{H_{2}}$ during β irradiation at room temperature was of 4 × 10^−7^ mol·J^−1^ at low dose level and only 2 × 10^−7^ mol·J^−1^ at about 10 MGy. Another example is reported in [30] where $G_{H_{2}}$ for the ultra-high molecular weight polyethylene (UHMWPE) during β irradiation (dose rate of 5 kGy/h) in vacuum and room atmosphere was 4.0 × 10^−7^ mol·J^−1^ at 1kGy and only of 1.8 × 10^−7^ mol·J^−1^ at 2 MGy. Consequently, our transducer had to be used in the low dose range domain, that is up to 200 kGy.

The microelectromechanical transducers designed, fabricated and characterized in this study were irradiated at room temperature (25–30 °C). We chose to fill the cavity with a thin film of HDPE polymer encapsulated in a vacuum for monitoring γ radiation in nuclear facilities such as plants and particle accelerators. The dose rate in such environments range typically between a few kGy·h^−1^ and a few 100 kGy·h^−1^. These devices can be used up to 200 kGy, but preferably less to ascertain a constant hydrogen emission yield factor.

## 4. Working Principle of the Microelectromechanical Transducer

### 4.1. Polymer Outgassing

Among many other characteristic reactions, outgassing occurs when a polymer is irradiated by γ-rays. To evaluate the structural modifications of polymers caused by such ionizing radiation, we consider here the gas emission yield factor $G_{gas}$ (in mol·J^−1^) of polymers. The number $n_{gas}$ of gas (in moles per unit of energy) deposited in the polymer during irradiation is then given by:(1)$$n_{gas}=m_{PO}\cdot G_{gas}\cdot D$$
where $m_{PO}$ denotes the mass (in kg) of polymer and D designates the dose level (in Gy or J·kg^−1^) deposited in the polymer. For a polymer of volume $V_{PO}$ placed inside a sealed structure of volume $V_{cav}$ (see Figure 3), the pressure $P_{g}$ (in Pa) generated by the radiolysis can be estimated from the ideal gas law as follows:(2)$$P_{g}=\frac{m_{PO}\cdot G_{gas}\cdot D\cdot R\cdot T}{V_{g}}$$
where R (=8.314 J·mol^−1^·K^−1^) is the ideal gas constant, T is the temperature (in K) and $V_{g}$ is the volume occupied by the gas (in m^3^) given by:(3)$$V_{g}=V_{cav}-V_{PO}$$

As can be derived from Figure 3, the cavity volume $V_{cav}$ is given by:(4)$$V_{cav}=V_{cav-glass}+V_{cav-Si}=V_{cav1-glass}+V_{ch}+V_{cav2-glass}+V_{cav-Si}$$
where $V_{cav-glass}$ and $V_{cav-Si}$ are the volumes of regions patterned in the glass and silicon substrates, respectively. Following (4), these volumes are composed of three different contributions: (i) the volume $V_{cav1-glass}$ of the cavity located below the membrane, (ii) the volume $V_{cav-PO}\;\left(=V_{cav2-glass}+V_{cav-Si}\right)$ of the cavity in which the polymer is placed, and (iii) the volume $V_{ch}$ of the channel between the two cavities. Moreover, the mass of polymer in (2) is given by:(5)$$m_{PO}=\rho_{PO}\cdot V_{PO}$$
where *ρ_PO_* designates the polymer density (in kg·m^−3^). Thus, the pressure $P_{g}$ (in Pa) generated by the radiolysis given by (2) can be rewritten as follows:(6)$$P_{g}=\rho_{PO}\cdot R\cdot T\cdot \frac{V_{PO}}{V_{g}}\cdot G\cdot D$$

### 4.2. Membrane Deflection

Let $t_{m}$ and $r_{m}$ be respectively the thickness and radius of the circular membrane (note that the radius $r_{m}$ is identical to the radius $r_{cav1-glass}=\frac{\Phi_{cav1-glass}}{2}$ of the circular *cavity 1* placed below the membrane). For very small deflections compared with the membrane thickness, and assuming perfect membrane clamping, the largest variation $\Delta W_{g}$ of the membrane deflection is given by the following approximation [78]:(7)$$\Delta W_{g}≅\;\frac{12}{64}\cdot \frac{1-\nu^{2}}{E}\cdot \frac{r_{m}{}^{4}}{t_{m}{}^{3}}\cdot P_{g}$$
where ν and E are, respectively, the Poisson coefficient and the Young’s modulus of the membrane. Furthermore, membrane deflection creates a volume variation which is negligible with respect to the overall cavity volume $V_{cav}$ The operating principle of the microelectromechanical transducer is based here on the measurement of the membrane deflection caused by the overpressure generated by the outgassing of an irradiated polymer. Hence, the relationship between the amount of gas produced by the irradiated polymer and the amplitude of the membrane deflection can be established. From (6) and (7), the descriptors of the two cavities, of the membrane and of the polymer, must be determined to control the amount of outgassing due to radiolysis and to maximize the membrane deflection.

## 5. Design Requirements of the Microelectromechanical Transducer

### 5.1. Requirements for Silicon Membrane Dimensions

According to (7) the thickness $t_{m}$ and radius $r_{m}$ of the silicon circular membrane have a significant impact on the largest variation $\Delta W_{g}$ of the membrane deflection or, equivalently, on the membrane sensitivity to applied pressure. The lower the thickness and larger the radius, the higher the sensitivity to applied pressure. However, membrane thickness and radius play a crucial role in the maximal stress $\sigma_{max}$ applied to the membrane, which may cause a fracture if it exceeds the membrane tensile strength. For a circular membrane clamped on the border, the approximation of maximal stress $\sigma_{max}$ can be derived from the following relationship [78]:(8)$$\sigma_{max}=\frac{3}{4}\cdot \Delta P\cdot {\left(\frac{r_{m}}{t_{m}}\right)}^{2}$$
where ΔP denotes the pressure applied to the membrane surface. Crystallographic orientation, shape and thickness of the monocrystalline silicon substrate have a strong impact on the membrane tensile strength [79,80,81,82,83,84,85,86,87,88,89,90,91]. However, to the best knowledge of the authors, few studies on the maximal tensile strength for thin film circular silicon membranes have been reported to date. The value of tensile strength for the silicon membrane was fixed here to 300 MPa, that is, close to the smallest tensile strength reported in scientific literature [90]. Consequently, the membrane dimensions were selected carefully in order to guarantee that the membrane stress $\sigma_{max}$ was lower than the maximal tensile stress of 300 MPa for applied pressure ΔP of 1 bar in standard conditions (that is, for a cavity in vacuum and the atmospheric pressure applied to the structure). It can be derived from Equation (8) that the following requirement must be fulfilled:(9)$$\frac{r_{m}}{t_{m}}<63\cdot 25$$

Moreover, the high voltage U (in V) involved during the anodic bonding process creates the pressure $P_{e}$ (in Pa) which causes the deflection of the membrane in the direction of the *cavity 1*. The pressure is given by [47]:(10)$$P_{e}=\frac{1}{2}\cdot \varepsilon_{0}\cdot {\left(\frac{U}{h_{cav-glass}}\right)}^{2}$$
where *ε_0_* is the permittivity of the vacuum (=8.85 × 10^−12^ F·m^−1^). The pressure $P_{e}$ will provide a requirement on the minimum value of the cavity height $h_{cav-glass}$ for avoiding undesirable bonding between the silicon membrane and the bottom of the cavity. To prevent such necking, the deflection $W_{e}$ due to the electrostatic pressure must be smaller than $\frac{2}{3}h_{cav-glass}$. Finally, the maximum deflection of the membrane W must be smaller than the height of the glass cavity $h_{cav-glass}$ at the initial state for which no irradiation occurs. Therefore, the following requirement must be fulfilled:(11)$$\frac{r_{m}{}^{4}}{t_{m}{}^{3}}<\frac{h_{cav-glass}}{\Delta P}\cdot \frac{64}{12}\cdot \frac{E}{1-\nu^{2}}$$

### 5.2. Requirements for the Cavities Dimensions

According to (6), the ratio $V_{PO}/V_{g}$ has a strong impact on the pressure $P_{g}$ due to radiolysis: the higher the ratio, the larger the pressure inside the structure. In order to ensure a large pressure $P_{g}$, we needed to use a high quantity of polymer (i.e., large volume $V_{PO}$) and a low volume containing the gas (i.e., low volume $V_{gas}$ or equivalently, low $V_{cav}$). As can be observed from Figure 3, $V_{PO}$ and $V_{cav}$ depend on the dimensions of cavities and the polymer. Since the polymer is placed inside the silicon cavity, a high ratio $V_{PO}/V_{g}$ requires a small contribution of the cavity etched in the glass substrate with respect to the total cavity volume (i.e., $V_{cav-glass}<10\%\cdot V_{cav}$). Assuming a negligible impact of $V_{cav1-glass}$ and $V_{ch}$ compared with that of the volume $V_{cav2-glass}$, high ratios of $V_{PO}/V_{g}$ are achieved when the following requirement is fulfilled:(12)$$h_{cav-glass}<\frac{1}{9}\cdot h_{cav-Si}\cdot {\left(\frac{\phi_{cav-Si}}{\phi_{cav2-glass}}\right)}^{2}$$

A good trade-off must be found between the dimensions of the membranes and cavities in order to achieve the reliability of the technological process, the high sensitivity of the membrane to applied pressure and sufficient mechanical robustness of the device.

### 5.3. Design Selection

The dimensions of cavities are reported in Figure 4 and Table 1 and Table 2. The diameter of the membrane cavity was set to 550 µm, while the membrane thickness *t_m_* was 5 µm and the cavity height $h_{cav-glass}$ 10 µm. Since the silicon membrane deflects in the crystallographic direction <001>, the Poisson coefficient ν and the Young’s modulus E of this membrane were set to 0.28 and 130 GPa, respectively. Following (7) the membrane sensitivity $\Delta W_{g}/P_{g}$ to applied pressure was found to be of 0.61 µm per 100 mbar. Furthermore, setting $h_{cav-glass}$ at 10 µm avoided the risk of unwanted bonding between the membrane and the cavity during the sealing process for applied voltages ranging from 0.5 kV to 1 kV, that is, for the voltage range used in [48] to obtain strong interface bonding. As sketched in Figure 4a,b, PE cavities in silicon and glass are designed with respective diameters of 5 and 6 mm. Since it is placed inside the silicon cavity, the PE sample must have a volume smaller than the volume of the cavity. Moreover, particular attention must be paid on the choice of the PE film thickness $t_{PE}$ and height $h_{cav-Si}$ of the silicon cavity in order to maximize the volume ratio $V_{PO}/V_{g}$ and, consequently, to maximize the applied pressure $P_{g}$ (see [6]). Two thicknesses $t_{PE}$ of HDPE films were selected for the analysis: $t_{PE}$ = 10 µm (commercially available [74]) and $t_{PE}$ = 200 µm (commercially available [92]). The impact of these thicknesses on the transducer performance was investigated when the cavities were filled with HDPE films. The corresponding ratio $V_{PO}/V_{g}$ is reported in Table 2 for a gap of 50 µm between the PE sample and the top of the silicon cavity. The lateral dimensions are given in Figure 4 and Table 1. For a HDPE sample having a disk shape and diameter of 3.5 mm, $V_{PO}/V_{g}$ was 7.02% for $t_{PE}$ = 10 µm and of 58.7% for $t_{PE}\;$= 200 µm. The resulting membrane sensitivity to applied pressure for $t_{PE}$ = 200 µm was eight times larger than that obtained for $t_{PE}$ = 10 µm. Consequently, the 200 µm thick PE sample was chosen for the fabrication of the MEMS transducer. Moreover, the PE filling of the cavity had to be analysed in order to understand polymer behaviour during the anodic bonding process, and to determine the most appropriate height of the cavity.

## 6. Manufacturing Process of the Transducer

### 6.1. Fabrication Process of the Microelectromechanical Transducer

Two substrates were required for the fabrication of the transducer, as sketched in Figure 3. Borosilicate glass 500 µm thick (B33 from Schott [63]) was used for the resonator substrate and was manufactured as sketched in Figure 5a (see details in [9]). After cleaning for 2 min in a bath in Piranha, and exposure to oxygen plasma for 10 min, the cavities and the channel shown in Figure 3 were fabricated. A 100 nm thick layer of chromium (Cr) and gold (Au) was first deposited on the surface through PVD deposition, and was patterned by a photolithography step and wet etching. The Cr/Au layer served as the hard mask for the wet etching process of the glass in a HF solution concentrated at 12.5%. Finally, both photoresist and metal layers were removed. The substrate dedicated to the membrane substrate fabrication was processed in parallel on a silicon-to-insulator (SOI) wafer from BT electronics [62], as illustrated in Figure 5b. The SOI substrate was composed of the 5 µm thick top (100) silicon layer and the 2 µm thick SiO_2_ layer above the 400 µm thick silicon bulk. Such a substrate is preferred to the standard Si wafer for better technological reliability of the membrane thickness. Any lack of control on this thickness would actually lead to unpredictable membrane sensitivity to applied pressure. First, the membrane substrate was cleaned in a bath with a 5% HF concentrated solution in order to remove the native oxide of the silicon. As the process was applied to the two substrate sides, alignment marks were necessary and were first patterned on the top face through photolithography and reactive ion etching (RIE). Next, photolithography and deep reactive ion etching (DRIE) under SF_6_ plasma were performed on the silicon bulk part until reaching the oxide layer, which acted as the etch stop layer. This layer was then removed using a wet etching process with a 5% HF concentrated solution. After the photoresist was removed through oxygen plasma, and after cleaning of the substrate as described above, the front side was processed using the photolithography step and DRIE under SF_6_ plasma to pattern the polymer cavity. After processing all the substrates, the anodic bonding was finally launched (see Figure 5c). A cleaning step of these substrates was first required. It consisted of a succession of different baths in order to remove remaining organic particles, native oxides and residues due to the different etching performed on the SOI and glass bonding surfaces: (i) the Piranha solution; (ii) a 5% HF concentrated solution; (iii) ammonia solution; (iv) deionised (DI) water rinsing and finally (v) megasonic bath immersion (frequency ranging from between 1 MHz to 100 MHz). The same process was applied to the glass substrate excluding the HF solution bath.

After the manufacturing process, the PE sample was introduced inside the silicon cavity etched from the top side and the different substrates were placed inside the wafer bonder. Following [48,53], the strong bonding interface was achieved here with a 400 °C temperature and a 1kV applied voltage. However, these conditions had to ensure the airtightness of the cavity without damaging the polymer. Therefore, the behaviour before irradiation of the polymer at high temperature and under a high applied voltage wase analysed.

### 6.2. Polyethylene Manufacturing

The technological processing of the PE thin film was required before filling the cavity. It consisted of several steps illustrated in Figure 6 and detailed below.

(i)A cleaning (step 1) using two different solvents (acetone and ethanol) in order to remove eventual organic pollution.(ii)Thermo-compression annealing in a vacuum by mechanically applying a pressure of 4 bar (step 2) to limit any structural deformation due to the high temperature of the anodic bonding.(iii)Shaping of the PE film (step 3) into a disk of 3.5 mm diameter by using a precise die cutter for good reproducibility of the polymer volume and mass, and the deposition of the polymer inside the cavity (step 4).(iv)The second annealing (step 5) was performed in a vacuum during the anodic bonding process for at least 30 min and before applying high voltage (step 6). During annealing, the glass and silicon substrates were separated to release the eventual polluting gas inside the polymer and the cavity.

The temperatures of annealing in steps 2 and 5 must were chosen to prevent the thermal degradation of the polymer, which could impact radiolysis and, consequently, could change the gas emission yield factor $G_{gas}$ of the PE. The appropriate temperature range was derived from the thermo-gravimetric analysis (TGA, see Section 7.1.1), which consisted of measuring the degradation of the polymer as the temperature increases. Structural studies also needed to be performed in order to estimate the impact of the second annealing process (step 5) and applied voltage (step 6) on the PE sample inside the cavity. Finally, the quantitative and qualitative gas composition was measured through high-resolution gas mass spectrometry of the polymer after γ-irradiation at different dose levels.

## 7. Physical Characterization of the Polyethylene

### 7.1. Characterisation of the Polymer before Irradiation

#### 7.1.1. Thermo-Gravimetric Analysis

This analysis (performed in ELEMCA laboratory, Toulouse, France) consists of measuring the mass variation of the PE material with the temperature and annealing duration. It allows the determination of the temperature at which the polymer degradation occurs, that is, the temperature at which the PE starts losing mass. The study was performed on a 15.2 mg HDPE sample under a nitrogen environment with a temperature increase of +5 °C·min^−1^ between 25 °C and 500 °C. The mass variation of this polymer is shown in Figure 7. A mass reduction of 5% (corresponding to a mass of 14.4 mg) occurred at 438 °C. From these points, a fast degradation occurred when the temperature increased. This result is in good agreement with those reported in the literature [21,66,67,68], showing a thermal stability up to 250–290 °C in air and about 400 °C in an inert atmosphere (vacuum, nitrogen or argon). The temperature used during the annealing processes was set here to 350 °C and constituted a good trade-off between PE preservation and bonding interface strength. At this temperature, no mass loss was observed in inert atmosphere. Besides, the temperature of the first annealing process (step 2) was set at 200 °C and corresponded to the highest temperature available from the used thermo-compression tool, but it remained significantly lower than the temperature degradation of PE under air.

#### 7.1.2. Microstructural Analysis

##### 7.1.2.1. PE Disk Thickness before the First Thermo-Compression Annealing

In order to analyse the uniformity of the PE thickness, tomographic measurement on three different HDPE disks was performed. Figure 8a displays thickness mapping of the disks surface, from which was derived the number of pixels for the thickness to take a predefined threshold (Figure 8b). The mean value of the thickness was then calculated for each disk as reported in Table 3. The global thickness average was 0.201 mm for HDPE disks, with a standard deviation of 4.2%. These results showed good technological reproducibility of the PE thickness.

##### 7.1.2.2. Structural Modification after Annealing in the Wafer Bonder

Morphological deformations of HDPE disks were investigated at different technological steps of the fabrication process. As shown in Figure 9a, HDPE samples were placed at the centre of the cavities etched in silicon, as shown in Figure 5b. The gap length (denoted by $D_{g}$) between the top of the HDPE sample and the top of the cavities were measured using an optical profilometer and cavity microsections (Figure 9b). These microsections were performed after cleavage of the substrate near the cavity edges. The structure was then coated with a resist to keep its integrity during final polishing, until reaching the cavity centre. The cavities patterned in silicon (diameter of 5 mm and height of 290 µm) were manufactured according to the process described in Section 6.1 on a standard silicon substrate. All PE disks used in this study were processed by following the technological steps detailed in Section 6.2 and are illustrated in Figure 6. Figure 10 shows microsection photographs after annealing at 350 °C in a vacuum (about 10^−4^ mbar) for 30 min in the wafer bonder. This step is equivalent to step 5 described in Section 6.2. Different parameters measured by using optical profilometer are reported in Table 4: (i) the gap length $D_{g}$; (ii) the maximum thickness $H_{MAX}$ of the HDPE samples; (iii) the maximum thickness variation $\Delta H_{MAX}$ of PE sample before and after the annealing and (iv) the diameter $\Phi_{PE}$ after annealing estimated from the microsection photographs in Figure 10. These pictures show modification of the disk shape from a flat into a convex profile. Furthermore, the disk diameter was not significantly modified by the annealing process (maximum diameter variation of 9%). The thickness (about 250 µm) of the HDPE samples was significantly modified during the process (+122% of the maximum thickness $H_{MAX}$) and, as reported in Table 4, exceeded the cavity height for all samples. This thickness modification may be an issue for PE sealing and, consequently, an appropriate encapsulation technique should be applied in future experiments to insert the polymer inside the cavity. The next section presents the properties of the different HDPE samples after annealing and application of the voltage during the bonding between silicon and glass substrates.

##### 7.1.2.3. Microstructural Study after the Application of the Voltage

Other cavities with 5 mm diameter and 320 µm height were fabricated on a standard silicon substrate and filled with the HDPE samples processed as detailed in Section 6.2. These samples were placed at the centre of the cavities. After annealing at 350 °C in a vacuum (at a pressure of about 10^−4^ mbar) for almost 1 h in the wafer bonder, the silicon substrate was bonded with the glass substrate applying a 600 V voltage at 350 °C in a vacuum. A contact force of about 100 N was applied between the silicon and glass substrates during voltage application. The cavities filled with PE before and after anodic bonding are shown in Figure 11a,b, respectively. As expected, it appears from Figure 11b and the microsection of the cavities shown in Figure 12 that all HDPE samples were present and concentrated at the centre of the cavity without impacting the overall bonding. The bonding interface presented a dark grey aspect without any colour fringes, as observed in [53]. This reflects good quality of the sealing. Although the thickness of PE samples was higher than the height of cavity after annealing (see Section 7.1.2.2), these results validated the proposed approach for the polymer encapsulation and the technological process for the fabrication of the MEMS transducer.

### 7.2. Irradiation of the PE Samples

To determine the gas and hydrogen emission yield factors $G_{gas}$ and $G_{H_{2}}$ of HDPE samples, high-resolution gas mass spectrometry was performed on each sample after irradiation. This technique allows the qualitative and quantitative measurement of the gas composition released during polymer radiolysis. The irradiations were performed using a Gammacell irradiator using ^137^Cs as the γ-ray source (energy of 6671.7 keV) in the CEA Saclay laboratory (Saclay, France). The PE sample was first cleaned and annealed at high temperature (below 350 °C) in a vacuum to remove eventual pollution inside the polymer. A sample of known mass was placed inside a 10 cm^3^ glass vial closed by a valve. The tightness was guaranteed by using an O-ring. Before closing the valve, three cycles of pumping in a vacuum and in helium (He) filling the vial were performed in order to remove the presence of any residual air. After these steps, the sample was placed inside the irradiator. The radiolysis of the polymer was performed inside the vial at room temperature and in a He inert atmosphere.

### 7.3. Identification of the Gas Released by the Polymer

After irradiation, the vial was connected to a high-resolution gas mass spectrometer. When the vacuum was achieved inside the measurement apparatus, the valve of the vial was opened and the gas was injected in the mass spectrometer [33], allowing the determination of the gas composition in the irradiated vial. Since the irradiation was performed in the He inert atmosphere, the quantity of helium was measured (helium occupied more than 99% of the total gas at the end of the experiment) and allowed the determination of the final pressure inside the vial, after irradiation. As described in the next section, the vial volume, which contained the polymer sample, was measured for determining the total gas and hydrogen emission yield factors in the He inert atmosphere. For the sake of illustration, Figure 13 displays the gas composition of a 49.7 mg HDPE sample placed inside a vial (volume of 9.84 cm^−3^), which had been irradiated at a dose of 24.03 kGy using a dose rate of 0.27 kGy·h^−1^. It shows the presence of $H_{2}$, hydrocarbon gas (such as $CH_{4}$, $C_{2}H_{4}$ and $C_{2}H_{6}$), but also of dinitrogen $N_{2}$ (41.6%), oxygen $O_{2}$ (0.6%) and water vapour $H_{2}O$ (2%). These gases were produced by the irradiated polymer or by the traces of residual air remaining in the vial after its conditioning. The lower oxygen quantity compared with the dinitrogen quantity can be explained by oxygen consumption of the polymer during radiolysis, which led to the presence of CO (6.2%) and $CO_{2}$ (7.4%). It may also have originated from a gas leak occurring during gas analysis with the mass spectrometer. To check the absence of reaction of the surrounding gas during irradiation, a glass container with gas but without polymer was irradiated with an identical dose rate and dose level. As expected (see Figure 14b), the concentration of gas constituents was almost the same before and after the irradiation. Both the eventual presence of $H_{2}$ in a quantity lower than the threshold detection, and the absence of $CH_{4}$, $C_{2}H_{4}$ and $C_{2}H_{6}$, confirmed that dihydrogen and hydrocarbon gases originated from polymer radiolysis. If these gases are removed from the PE composition histogram (see Figure 14a), the dinitrogen concentration is found to be very close to the one obtained from the empty test vial and PE samples. Furthermore, the ratio between dinitrogen and dioxygen concentrations derived from the empty vial histogram (Figure 14b) was about 2.5 (with and without irradiation) and was found to be lower than the ratio (=3.7, see Table 5) measured in standard atmosphere. Besides, the water vapour, dinitrogen and argon concentrations were almost the same for the HDPE analysis and the reference test case. Consequently, these gases did not originate from any leakage during vial conditioning or from polymer outgassing, but were released during experimental gas mass analysis. Moreover, the oxygen consumption by the polymer and its conversion into carbon monoxide and dioxide during irradiation were confirmed by comparing their respective concentrations between the irradiated HDPE (case A) and empty vial (case B). By taking into account the presence of CO, $CO_{2}$ and $O_{2}$, a total volume proportion of about 23.5% in case A was obtained, which was close to the proportion of $O_{2}$ (27.5%) in case B. Consequently, the analysis of the gas composition due to PE radiolysis required removing the contribution of the following gases: He (surrounding atmosphere during the irradiation), $O_{2},$ CO, $CO_{2},$ Ar, $N_{2}$ and $H_{2}O$. The result of this analysis is reported in Figure 15. A hydrogen concentration of about 89.8% and a proportion of 5.1% of hydrocarbon gases were finally obtained, which was in good agreement with state-of-the-art results reported in Section 3.2. From these measured proportions, the gas emission yield factor of HDPE samples is derived in the next section.

The emission yield factor $G_{x}$ of gas constituent x can be estimated as follows:(13)$$G_{x}=\frac{{\%}_{x}^{vol}\cdot P_{f}\cdot V_{vial}}{m_{PO}\cdot R\cdot T\cdot D}$$
where $V_{vial}$ is the free volume of the vial (in m^3^), $P_{f}$ denotes the pressure inside the vial after irradiation (in Pa)*,*
${\%}_{x}^{vol}$ designates the volume fraction of gas constituent x, and D the dose level (in Gy). From Figure 13 and Figure 15, the emission yield factors were determined and are reported in Table 6. As expected (see Section 3.2), the major contribution to gas generation in the radiolysis process was that of hydrogen, with a contribution of 90%. In Table 7 are reported some hydrogen emission yield factors for various film thicknesses and two dose levels (12.15 kGy and 24.03 kGy). This factor appeared not to be significantly dependent on the film thickness and dose level, at least for doses lower than 24.03 kGy. The maximum duration time $\Delta t_{MAX}$ (in s) necessary for hydrogen to diffuse outside of the polymer can be estimated from the Fick’s law [76] as follows:(14)$$\Delta t_{MAX}=\frac{h_{HDPE}^{2}}{6D_{H_{2}}^{HDPE}}$$
where $h_{HDPE}$ (in µm) is the film thickness and $D_{H_{2}}^{HDPE}$ (in µm^2^·s^−1^) denotes the hydrogen diffusion coefficient in the polymer. At a film thickness ranging from 10 µm to 1000 µm and a typical diffusion coefficient of 2.2 × 10^6^ cm^2^·s^−1^ [76], $\Delta t_{MAX}$ was between about 2 s and 13 min, that is, a duration shorter than the time between the end of the irradiation and beginning of the gas composition analysis. This confirms that there was no residual gas trapped in the HDPE film during our experiment and, consequently, the total quantity of hydrogen produced during the radiolysis was accurately estimated. From (13), the measurement uncertainty on gas emission yield factor $G_{x}$ can be determined as follows:(15)$$\frac{\Delta G_{x}}{G_{x}}=\sqrt{{\left(\frac{\Delta {\%}_{x}^{vol}}{{\%}_{x}^{vol}}\right)}^{2}+{\left(\frac{\Delta P_{f}}{P_{f}}\right)}^{2}+{\left(\frac{\Delta V_{vial}}{V_{vial}}\right)}^{2}+{\left(\frac{\Delta m_{PE}}{m_{PE}}\right)}^{2}+{\left(\frac{\Delta T}{T}\right)}^{2}+{\left(\frac{\Delta D}{D}\right)}^{2}}$$
where ΔX (for $X={\%}_{x}^{vol},P_{f},\;V_{vial},\;m_{PE},\;T,\;D)$ denotes the absolute standard deviation on X values (see Table 8). The measurement uncertainty on the emission yield factor of each gas constituent is ±10.4%.

## 8. Theoretical Estimations and Measurement Uncertainties

### 8.1. Validation of the Anodic Bonding Conditions and the Selected Dimensions

Analysis of the PE material exposed as described in the previous sections allowed determination of the anodic bonding conditions, that is, the temperature of 350 °C under vacuum, the applied voltage of 600 V and the contact force of about 100 N between the glass and silicon substrates. While keeping polymer integrity, the assembling step performed under vacuum would also prevent any effect of the silicon membrane roughness on the deflection measurement. According to (7), bonding under a vacuum condition provides larger membrane displacement than bonding under atmospheric pressure: the pressure gradient is actually about 1bar if the anodic bonding is performed in vacuum, while it is only of a few mbars when bonding is done under atmospheric pressure. Furthermore, the selected dimensions of the structure (Table 1 and Table 2) combined with a pressure gradient in the vacuum anodic bonding avoided the two following issues.

(i) The membrane breaking, as the maximal stress (about 240 MPa according to (8)) was smaller than the membrane tensile strength (300 MPa).

(ii) Membrane collapse, as the maximum membrane deflection (6.83 µm) was lower than the height (10 µm) of the cavity. By assuming a flat displacement of the membrane, this corresponds to a volume variation of around 4.60 × 10^−4^ mm^3^ which can be neglected with respect to a glass cavity volume $V_{cav-glass}$ of 2.86 × 10^−1^ mm^3^ (see Table 1).

The final dimensions of the dosimeters are reported in Figure 16, which shows the PE cavity (height of 290 µm) etched in silicon substrate.

### 8.2. Measurement Uncertainty on the Pressure Generated by the Radiolysis

From (2), the following measurement uncertainty on the pressure $P_{g}$ generated by the radiolysis can be determined:(16)$$\frac{\Delta P_{g}}{P_{g}}=\sqrt{{\left[\frac{\Delta T}{T}\right]}^{2}+{\left[\frac{\Delta G_{gas}}{G_{gas}}\right]}^{2}+{\left[\frac{\Delta m_{PE}\;}{m_{PE}}\right]}^{2}+{\left[\frac{\Delta V_{g}}{V_{g}}\right]}^{2}+{\left[\frac{\Delta D}{D}\right]}^{2}\;}$$

Section 8.2.1 is devoted to the detailed calculation of $\frac{\Delta m_{PE}\;}{m_{PE}}$, $\frac{\Delta V_{g}}{V_{g}}$ is determined in Section 8.2.2, and finally, the measurement uncertainty $\frac{\Delta P_{g}}{P_{g}}$ on the pressure $P_{g}$ generated by the radiolysis is derived in Section 8.2.3.

#### 8.2.1. Estimation of the Mass Measurement Uncertainty $\frac{\Delta m_{PE}}{m_{PE}}$

Mass uncertainty was first estimated from the measurement of several PE samples. These samples were shaped from a HDPE film and processed by following the steps 1 to 3 detailed in Section 6.2. The mass measurement was performed using a microbalance Mettler Toledo XSE [93] with a precision of ±0.01 mg. From 20 mass measurement results of the same sample, it was derived that the measurement uncertainty of mass was ±2.3% (see Table 9). Next, the variability analysis on the shaping process of PE disk samples was performed by measuring the mass of 35 HDPE disk samples. Both average and standard (absolute and relative) deviations are reported in Table 10. The average mass of a processed disk sample was 1.78 mg with a mass measurement uncertainty $\frac{\Delta m_{PE}\;}{m_{PE}}$ of 5.1%.

#### 8.2.2. Estimation of the Uncertainty Measurement $\frac{\Delta V_{g}}{V_{g}}$

The absolute uncertainty measurement of the volume $V_{g}$ can be derived from (3) and (4).
(17)$$\frac{\Delta V_{g}}{V_{g}}=\sqrt{{\left[\frac{\Delta V_{cav-glass}}{V_{cav-glass}}\right]}^{2}+{\left[\frac{\Delta V_{cav-Si}}{V_{cav-Si}}\right]}^{2}+{\left[\frac{\Delta V_{PE}}{V_{PE}}\right]}^{2}}$$

In (17), $\Delta V_{PE}$ was determined from the results reported in Section 8.2.1. The analysis of the technological process of glass and silicon substrates was reported for estimating $\Delta V_{cav-glass}$ and $\Delta V_{cav-Si}$, and finally the absolute uncertainty measurement of the volume $V_{g}$ was derived from (17).

From the definition of mass density, one can establish that the uncertainty $\frac{\Delta V_{PE}}{V_{PE}}$ on the PE volume is given by:(18)$$\frac{\Delta V_{PE}}{V_{PE}}=\frac{\Delta m_{PE}}{m_{PE}}$$

For the 3.5 mm diameter of the disk sample, and using the average thickness of the disk determined in Section 8.2.1 and the mass uncertainty reported in Table 10, the value was $\Delta V_{PE}=0.098{\;\mathrm{mm}}^{3}$. Moreover, the photolithography process (see step 3 in Figure 5a and step 9 in Figure 5b) included photoresist coating, exposition and development, which generated measurement uncertainty on the cavity dimensions due to apparatus inaccuracies and human operator interventions during the technological process. The maximum absolute uncertainty $\Delta z_{max}$ on each cavity dimension due to the photolithography process performed on glass and SOI substrates was 2 µm. As shown in Figure 4b and detailed in Section 5.3, the cavity fabricated on the glass substrate was composed of three parts: two circular cavities (volumes $V_{cav1-glass}$ and $V_{cav2-glass}$) linked by a channel (volume $V_{ch}$). From Table 1, it can be observed that $V_{ch}$ and $V_{cav1-glass}$ represented only 1.2% of the total volume $V_{cav-glass}$. Consequently, the volume uncertainty $\frac{\Delta V_{cav-glass}}{V_{cav-glass}}$ in (17) is given as follows:(19)$$\frac{\Delta V_{cav-glass}}{V_{cav-glass}}≅\frac{\Delta V_{cav2-glass}}{V_{cav2-glass}}=\sqrt{4.{\left[\frac{{\Delta \Phi }_{cav2-glass}}{\Phi_{cav2-glass}}\right]}^{2}+{\left[\frac{{\Delta h}_{cav-glass}}{h_{cav-glass}}\right]}^{2}}$$

Therefore, the estimation of $\frac{\Delta V_{cav-glass}}{V_{cav-glass}}$ required the estimation of the variations in the glass cavity diameter $\Phi_{cav2-glass}$ and of the depth etching in glass $h_{cav-glass}$. As detailed in Section 6.1, two wet etchings were performed on the glass substrate. Consequently, over-etching may occur during the metal and glass etchings, which might impact on the final cavity dimensions. By taking into account the etching depth $h_{etch}$, the undesirable over-etching effect $x_{etch}$ can be estimated as follows [94]:(20)$$x_{etch}≅0.7h_{etch}$$

Photolithography adds the so-called *over-etched diameter* $\phi_{cav2}^{over}$ to the cavity 2, which is given by:(21)$$\phi_{cav2}^{over}=\phi_{cav2}^{th}+\Delta z_{max}+2\left(x_{metal}+x_{glass}\right)=\phi_{cav2}^{th}+\Delta z_{max}+1.4\left(h_{metal}+h_{cav-glass}\right)$$
where $x_{glass}$ denotes the over-etching impact on the thickness ($h_{cav-glass}$) of the glass substrate, $x_{metal}$ denotes the over-etching impact on the thickness ($h_{metal}$) of the metal layer, and $\phi_{cav2}^{th}$ designates the targeted (or theoretical) diameter of the cavity. The thickness $h_{metal}$ of the metal layer was approximately of 100 nm (see Section 6.1) and was negligible compared with the etching depth (10 µm) performed on glass substrate. Therefore, $\Phi_{cav2-glass}≅\phi_{cav2}^{over}$ and ${\Delta \Phi }_{cav2-glass}≅\Delta \phi_{cav2}^{over}$. The absolute uncertainty $\Delta \phi_{cav2}^{over}$ is then given as follows:(22)$$\Delta \phi_{cav2}^{over}=\Delta z_{max}+1.4\;\Delta h_{cav-glass}$$

In order to evaluate the maximum uncertainty $\Delta h_{cav-glass}$, a reproducibility study was performed by measuring the depth etching $h_{cav-glass}$ of 64 different cavities using a mechanical profilometer. The standard deviation, average and relative uncertainties are reported in Table 11. They were determined using four glass substrates with 16 cavities with a targeted height of 9 µm (this height was obtained from a 25 min bath in a 12.5% concentrated HF solution). The reproducibility of the glass etching led to a variation of 1.0% on the cavity height. It was then possible to determine the relative uncertainty on the cavity volume etched into the glass substrate.

Table 12 indicates that this uncertainty was 0.9% for a glass cavity height $h_{cav-glass}$ of 10 µm. We then proceeded to estimate $\frac{\Delta V_{cav-Si}}{V_{cav-Si}}$ in (17). As it can be observed from Figure 4a, only the cavity of volume $V_{cav-Si}$ had an impact on the volume occupied by the gas. The uncertainty $\frac{\Delta V_{cav-Si}}{V_{cav-Si}}$ is given by:(23)$$\frac{\Delta V_{cav-Si}}{V_{cav-Si}}=\sqrt{4.{\left[\frac{{\Delta \Phi }_{cav-Si}}{\Phi_{cav-Si}}\right]}^{2}+{\left[\frac{{\Delta h}_{cav-Si}}{h_{cav-Si}}\right]}^{2}}$$

Since the cavities were etched into the silicon substrate from DRIE (see Section 6.1), there was no over-etching effect during the technological process. Only photolithography could generate inaccuracy in the diameter $\Phi_{cav-Si}$ of the PE cavity given by:(24)$$\phi_{cav-Si}=\phi_{cav-Si}^{th}\pm \Delta z_{max}=\phi_{cav-Si}^{th}\pm \Delta \phi_{cav-Si}$$
where $\phi_{cav-Si}^{th}$ designates the targeted (or theoretical) diameter of the cavity and where $\Delta \phi_{cav-Si}$ denotes the uncertainty of the cavity diameter fabricated in the silicon. Following the above-described method, the technological reproducibility of the DRIE etching was analysed. The height measurement of 19 silicon cavities with height close to 300 µm was performed on four different silicon substrates. From Table 13 it can be determined that the reproducibility of the DRIE Si etching provided an uncertainty of 0.7% on the cavity height. Both relative and absolute uncertainties on the volume $V_{cav-Si}$ were finally calculated and are reported in Table 14.

From (17) and uncertainties reported in Table 12 and Table 14, both absolute uncertainty $\Delta V_{g}$ and relative uncertainty $\frac{\Delta V_{g}}{V_{g}}$ of the volume occupied by the gas in the dosimeter could finally be estimated (see Table 15). One finds $\frac{\Delta V_{g}}{V_{g}}=5.3\;\%$.

#### 8.2.3. Measurement Uncertainty $\frac{\Delta P_{g}}{P_{g}}$ on the Pressure $P_{g}$ Generated by Radiolysis

Table 16 reports the relative uncertainty $\frac{\Delta P_{g}}{P_{g}}$ on the pressure *P*_g_ generated during the radiolysis derived from (16) and from Section 8.2.1 (calculation of $\frac{\Delta m_{PE}\;}{m_{PE}}$) and Section 8.2.2 (calculation of $\frac{\Delta V_{g}}{V_{g}}$). The pressure uncertainty $\frac{\Delta P_{g}}{P_{g}}$ was 16.2%.

### 8.3. Theoretical Estimation of the Membrane Deflection

Table 17 reports the pressure $P_{g}$ obtained from (2). The gas emission yield factor $G_{gas}$ was set to 4.4 × 10^−7^ J·mol^−1^ (see Table 6) and the temperature T to 300 K. In this table, the membrane deflection $\Delta W_{g}$ ($\Delta W_{g}=0$ when there is no irradiation) is calculated from (7) and a membrane thickness $t_{m}$ of 6.64 µm. This value corresponds to the average of several thickness measurements performed by using focused ion beam (FIB) sections on dummy cells (see Figure 17).

## 9. Experimental Validation

### 9.1. Measurement Setup

As mentioned in Section 4.1, the quantity of gas produced during the polymer radiolysis inside the sealed cavity can be derived from the maximum deflection amplitude of the silicon membrane. For experimental validation purpose, this amplitude was measured from an interferometric profilometer, which provided a contactless scanning of the membrane surface. Data processing of the surface scanning allowed retrieval of the maximum deflection amplitude at the centre of the membrane with a measurement uncertainty of ±0.05 µm. On one hand, the deflection analysis allows verification of the long-term airtightness of the structure after irradiation (and for different dose levels) and validation of the choice of dosimeter materials and the assembling technique. On the other hand, the correlation between the pressure-dependent deflection amplitude with the gas emission yield factor of the polymer demonstrates the feasibility of the proposed dosimeter topology. Note that the deflection amplitude is negative when the membrane deflects in the direction of the cavity. This occurs when the pressure $P_{in}$ inside the sealed cavity is lower than the pressure $P_{out}$ outside the cavity (see Figure 18).

### 9.2. Long-Term Stability of Airtightness

The airtightness of different MEMS transducer was investigated by measuring variation of the deflection over time. The gas quantity used for obtaining the results reported in Figure 19 refers to the relative deflection normalized with respect to the first deflection measurement at t = 0. This analysis was performed on four cells (denoted #1 to #4) filled with polymer. All these measurements were made under an external room atmosphere. The samples were irradiated four times at the dose level of around 20 kGy and a dose rate of 0.36 kGy·h^−1^. The irradiation equipment was the same as described in Section 7.2. Each irradiation step is indicated by a coloured arrow in Figure 19. The airtightness of the MEMS transducers for a long period (at least 2 years and half) is sufficient for our application, as the relative deflection is constant between two irradiations and does not increase over time. Furthermore, we observed that the fabricated sensor can be used for measuring dose levels up to 80 kGy. This result validates experimentally the sensor design. Other dummy cells without polymer were also used to evaluate the integrity of the structure for high dose levels up to few MGy. The irradiations were performed at LABRA (CEA Saclay, Saclay, France) using ^60^Co (photon energy: 1.17 and 1.33 MeV) as the γ-ray source with a dose rate of 1.7 kGy·h^−1^. Three dose levels were used on the dummy structures: 501.2 kGy (applied on cell#5; #6; #10 and #11); 1104.1kGy (applied on cell#7) and 4032.0 kGy (applied on cell#8 and #9). From the graph presented in Figure 20, no deflection shift is observed for a long period (at least 2 years and half), proving the compatibility of the selected materials and assembling technique for the targeted application, and for a dose range up to at least 4 MGy.

### 9.3. Radiolysis Pressure $P_{g}$ Extraction

The variation of the silicon membrane deflection of cell#1 to #4 at different dose levels up to 80 kGy is displayed in Figure 21a–d. As a first approximation, the relationship between deflection and dose level was linear (the coefficient of determination of the linear regressions is at least of 0.93). For most membranes, the initial deflection before irradiation (at 0 kGy) was similar with a mean value of −5.3 µm and a relative standard deviation of 2.2%, while the membrane sensitivities to dose level were close: the mean value of this sensitivity was 14.65 nm·kGy^−1^ with a relative standard deviation of 11.1%. Moreover, the deflection measured before the first irradiation confirmed that the cavity was in a vacuum. Figure 22 displays the shift $\Delta W_{g}$ of the deflection shift before irradiation. This shift was computed from Table 17 and for various dose levels. The measured deflection shifts are in good agreement with the predicted values. Therefore, the proposed model can be used to estimate the pressure $P_{g}$ generated by the PE radiolysis for the different dose levels up to 80 kGy.

Another experimental set up was used to obtain more accurate estimations of the pressure $P_{g}$. The MEMS test devices were placed inside an airtight chamber where the pressure (noted $P_{out}$ in Figure 18) was monitored. The deflections were measured by an interferometric profilometer for different applied pressures. The technique consisted of achieving after each irradiation the deflection measured before the irradiation. The pressure $P_{g}$ to be applied to the membrane for retrieving the deflection before the very first irradiation can be predicted from:(25)$$P_{g}=P_{out}^{AI-k}-P_{out}^{NI}$$
where $P_{out}^{NI}$ and $P_{out}^{AI-k}$ are, respectively, the external pressure applied on the cell nonirradiated (*NI* state) and after the kth irradiation (*AI-k* state) and leading to the same deflection value in the *NI* and *AI-k* states. This method does not depend on the eventual pressure inside the sealed cavity before irradiation. Prior to the estimation of the pressure $P_{g}$, the mechanical properties of the membrane were first investigated. Figure 23 displays the membrane deflection W as a function of the external pressure $P_{out}$ applied to the different structures and before the first irradiation. The measurement uncertainty on this pressure was ±20 Pa. As a first approximation, the relationship between deflection and the applied pressure was linear (the coefficient of determination of the linear regressions was at least 0.97). The uncertainty δP on the applied pressure depends on the uncertainty δW on the membrane deflection as follows:(26)$$\delta P=\delta W/\left|\alpha \right|$$
where *α* denotes the mechanical sensitivity of the membrane linked to the deflection variation with respect to the applied pressure (in µm·bar^−1^). Estimation of δP from (26) was found to be of 11 mbar for all the passive dosimeters. The radiolysis pressure $P_{g}$ was derived from Figure 24 for all the fabricated cells. This figure displays the pressure per kGy for each dose level ranging from 20 kGy to 80 kGy.

The results reported in Figure 24 are in good agreement with the theoretical predictions up to 40 kGy (all the data points belong to the green region of the figure). The gas emission yield factor $G_{gas}$ was 4.4 × 10^−7^ mol·J^−1^ for dose levels ranging from 20 kGy to 80 kGy. All the experimental data points were below the expected value for doses between 60 kGy and 80 kGy. This can be explained by the reduction of the yield factor $G_{gas}$ compared with the factor at 40 kGy, which leads to a nonlinear relationship between the pressure $P_{g}$ and the dose level. A. Ventura et al. proposed in [95] to model the $H_{2}$ emission yield factor $G_{H_{2}}$ as follows:(27)GH2D=GH2sat1−1−GH2D→0GH2sat·e−γ·D
where D is the dose level, $G_{H_{2}}^{D\to 0}$ and $G_{H_{2}}^{sat}$ are $H_{2}$ emission yield factor, respectively, at low and high dose levels. According to (6), the applied pressure $P_{g}$ can be rewritten as follows:(28)Pg=ρPO·R·T·VPOVg· GH2sat1−1−GH2D→0GH2sat·e−γ·D·D

Figure 25 shows two different models used to describe the relationship between the average radiolysis pressure and the dose level: (a) a linear regression model and (b) an exponential law given by Equation (28). It appears that the exponential law is in better agreement with the experimental results than the linear relationship. Nevertheless, by taking into account the uncertainties determined in Section 8.2, the emission yield factor G can be assumed constant. As a matter of fact, the difference between the measured radiolysis pressure at 80 kGy and the expected pressure range (see Figure 24) was only about 5.3%. As the uncertainty δP on the applied pressure was ±11 mbar, the linear relationship between the average radiolysis pressure and the dose level would lead to a resolution of about 2.8 kGy for the proposed passive dosimeter. Note that the most accurate model, linear vs. exponential, for describing the relationship between the average radiolysis pressure and dose level should be selected from the measurement of many samples irradiated at a dose level of at least 40 kGy and from the analysis of gas composition using the mass spectrometry. This selection could be applied in a future investigation.

### 9.4. Detection Limit, Thermal Variation Impacts and Longtime Airtightness Analysis

The detection limit of the mechanical transducer were evaluated using the measurement uncertainty of the deflection (0.05 µm) and the linear regression between the deflection and the nuclear dose. According to the values obtained in Figure 21a–d, the minimum irradiation dose which detectable by the proposed devices was 3.4 kGy. However, as explained in Section 2 and depicted in Figure 1, the transducer was a first prototype to be included in the final RF dosimeter (under development). Nevertheless, the measurement resolution in the irradiation dose of the overall passive and wireless RF dosimeter can be estimated. The full-scale measurement range of both the final sensor and MEMS transducer was 1bar, as the mechanical structure could detect absolute pressure inside the cavity from 0 bar to 1 bar. Furthermore, the authors showed in [35] that a 50 mbar minimal detectable applied pressure (i.e., 5% of the full-scale measurement range) can be achieved from passive and wireless sensors. Consequently, with a radiolysis pressure $P_{g}$ of 4.92 mbar·kGy^−1^, the measurement resolution of the sensor was 10.2 kGy. Such a detection limit may be decreased from an optimized sensor design. The first solution consisted of improving the radiolysis yield of the polymer, which led to an increase of the *G* factor and also of the quantity of gas inside the cavity after irradiation. The second solution consisted of increasing the volume ratio $V_{PO}/V_{g}$ by reducing the volume etched in the silicon cavity. Considering the cavity with a depth of 210 µm (instead of 290 µm) and a diameter of 4.5 mm (instead of 5 mm), the $V_{PO}/V_{g}$ ratio was 1.13, i.e., 2.4 higher than the original value. The radiolysis pressure $P_{g}$ was then higher at almost of 12 mbar·kGy^−1^. At the same time, the measurement resolution of the irradiation dose decreased to 4.2 kGy.

Furthermore, even if not investigated in this paper, we can consider two main thermal phenomena which may impact the passive mechanical transducers characteristics.

(i)The effect of the temperature variation on the gas dilatation and pressure inside the cavity. Assuming a temperature variation ΔT of 30 °C around a room temperature of 25 °C, we can achieve a relative temperature variation (and so a relative pressure variation inside the cavity) of about 10%. It is possible to measure and to compensate, in part, for the effect of the pressure variation by using a passive temperature sensor [96].(ii)The effect of the temperature variations on the differential expansion behaviour between constitutive materials (i.e., glass and silicon) of the sensor, which may modify the mechanical properties of the membrane. However, this effect is assumed to be negligible because glass and silicon have similar linear thermal expansion coefficients over a wide temperature range. For example, for temperature ranging from −40 °C to +60 °C, the thermal expansion coefficient is 3.2 × 10^−6^ K^−1^ for the borosilicate material (see [63]) and between 2.0 × 10^−6^ K^−^^1^ and 2.9 × 10^−6^ K^−1^ for the silicon material (see [97]). Moreover, the silicon-glass technology has been successfully used for manufacturing capacitive pressure sensors with a temperature sensitivity of some 100 ppm·K^−1^ in this large temperature range [98].

Even if the airtightness of the structure was validated for at least 2 years and half (as shown in Figure 19), it could be interesting to evaluate the time necessary to detect a specific variation of pressure due to a leak in the device. In the context of nuclear dose monitoring in a standard environment (i.e., 25 °C and 1bar), and supposing a radiolysis pressure of 1bar inside the cavity, the gas composition and partial pressure for each component inside and outside the MEMS transducer are presented in Table 18. Furthermore, the permeation rates $K_{X}^{M}$ reported from different articles, as in [52,99,100,101], are tabulated for each gas component (noted X) through constitutive materials (noted M) at a given temperature. $\left|\Delta P_{X}\right|$ denoting the absolute value of the partial pressure difference for the gas component X inside and outside of the cavity. Strong permeation of a gas X through a material M depends on both a high partial pressure difference $\left|\Delta P_{X}\right|$ and a high permeation rate $K_{X}^{M}$. According to the $\left|\Delta P_{X}\right|$ values in Table 18, the main components susceptible to permeate trough the structures are the following: $N_{2}$; $O_{2}$; $H_{2}O$ and $H_{2}$. To the best knowledge of the authors, no permeation rate has been identified in the literature for $O_{2}$; $H_{2}O$; $CO_{2}$; $C_{x}H_{y}$ in silicon material, and for Ar; $H_{2}O$; $CO_{2}$ $C_{x}H_{y}$ in glass substrate. Since $K_{He}^{Si}$ corresponding to the largest permeation rate value found at room temperature is very small compared to $K_{He}^{SiO_{2}}$, the permeation phenomenon through silicon material for all gases can be considered negligible compared to the permeation through glass. Besides, the diffusion parameter of $H_{2}O$ in silicon dioxyde at room temperature presented in [102] and shown in Table 19 is almost 5.3 × 10^11^ smaller than the diffusion coefficient of He. Consequently, we may assume that water permeation through glass can be neglected. By simplification, only the permeation of $H_{2}$; $N_{2}$ and $O_{2}$ should be taken into account.

The permeation rate for each gas component allowed the determination of the leak rate $L_{X}^{SiO_{2}}$ for each molecule, as expressed in (29):(29)$$L_{X}^{SiO_{2}}=K_{X}^{SiO_{2}}\cdot \frac{S}{t_{SiO_{2}}}\cdot \left|\Delta P_{X}\right|$$
where S and $t_{SiO_{2}}$ represent respectively the area of the cavity from which the species goes inside and outside of the device (evaluated at around 1.1 cm^2^ by considering the dimensions of the MEMS transducer) and the thickness of the glass substrate after etching (taken at 490 µm). The values of the different leak rate for the selected component are given in Table 20.

From the leak rates, it was possible to calculate the time $T_{X}^{SiO_{2}}$ necessary to change the pressure inside the cavity of a value δP due to the entrance (in the case of $N_{2}$ and $O_{2}$) or the escape (here $H_{2}$) from the cavity by using the following expression [103]:(30)$$T_{X}^{SiO_{2}}=\frac{V_{g}}{L_{X}^{SiO_{2}}}\cdot \left|P_{X}^{out}-P_{X}^{in}\left(0\right)\right|\cdot ln\left(\left|\frac{P_{X}^{out}-P_{X}^{in}\left(0\right)}{P_{X}^{out}-P_{X}^{in}\left(T\right)}\right|\right)$$
where $P_{X}^{out}$ and $P_{X}^{in}$ correspond respectively to the partial pressure of the component X outside and inside of the cavity at an initial time *T* = 0 (noted $P_{X}^{in}\left(0\right)$) and at a time *T* (noted $P_{X}^{in}\left(T\right)=\left|P_{X}^{in}\left(0\right)–\delta P\right|$). Table 21 presents different values of time $T_{X}^{SiO_{2}}$ for each component and for different δP. From this table, we can see that the escape of $H_{2}$ from the cavity happens first, and that a shift δP of 1 mbar inside the cavity may occur in almost 20 years.

## 10. Conclusions

This paper reports the design, technological fabrication and characterization of a microelectromechanical transducer for monitoring irradiation dose level in nuclear environments. This device is composed of an airtight cavity containing a polymer (HDPE), which releases gases under nuclear radiation. The pressure generated by the polymer radiolysis during irradiation was estimated from measurement a membrane deflection. A disk of HDPE of 3.5 mm diameter and 200 µm thick was used. Good airtightness of the cavity was obtained for at least 2.5 years for a dose level of 4 MGy. The eventual impact on the HDPE disk of the anodic bonding used during the technological fabrication process was investigated. A maximum temperature of 350 °C was recommended to ensure both the strong bonding interface and no mass loss of the polymer. It was also observed that both temperature and voltage led to the deformation of the HDPE disk, but this did not damage the bonding interface. The nature and quantity of the gas released by irradiated HDPE film was measured by using a high-resolution gas mass spectrometry for dose levels up to 24 kGy. Both gas emission yield factor and gas composition were consistent with available published data. Next, an experimental setup was proposed to estimate the radiolysis pressure from the measurement of the membrane deflection. The sensor performances were validated experimentally up to 80 kGy and were in good agreement with theoretical predictions.

The impact on the sensor performances of α, β or neutron irradiations, the temperature and dose rate could be investigated in future work. Moreover, even if they do not alter the interface bonding, the observed deformations of the polymer due to irradiation may be reduced from polymer encapsulation before the sealing process. Since the ratio between the volume occupied by the gas inside the cavity and the volume of the polymer play a role in sensor sensitivity to dose level, the combination of encapsulation with the stacking of the thin polymer film may actually enhance the sensitivity of the MEMS dosimeter.

## Figures and Tables

**Figure 1 sensors-21-05912-f001:**
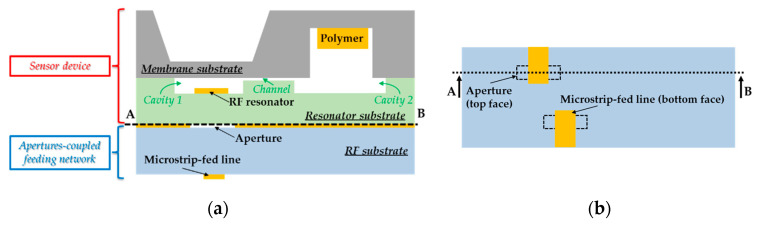
Cross-sectional view of (**a**) the proposed passive microelectromechanical dosimeter and (**b**) its bottom view showing the microstrip line used to feed the two-port planar resonator through two small apertures by electromagnetic waves (see [37,38]). (**c**) The mechanical MEMS transducer studied specifically in this paper.

**Figure 2 sensors-21-05912-f002:**
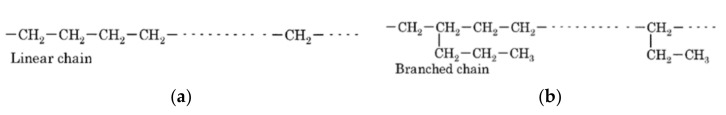
Molecular structure of (**a**) high-density polyethylene (HDPE) and (**b**) low-density polyethylene (LDPE) [30].

**Figure 3 sensors-21-05912-f003:**
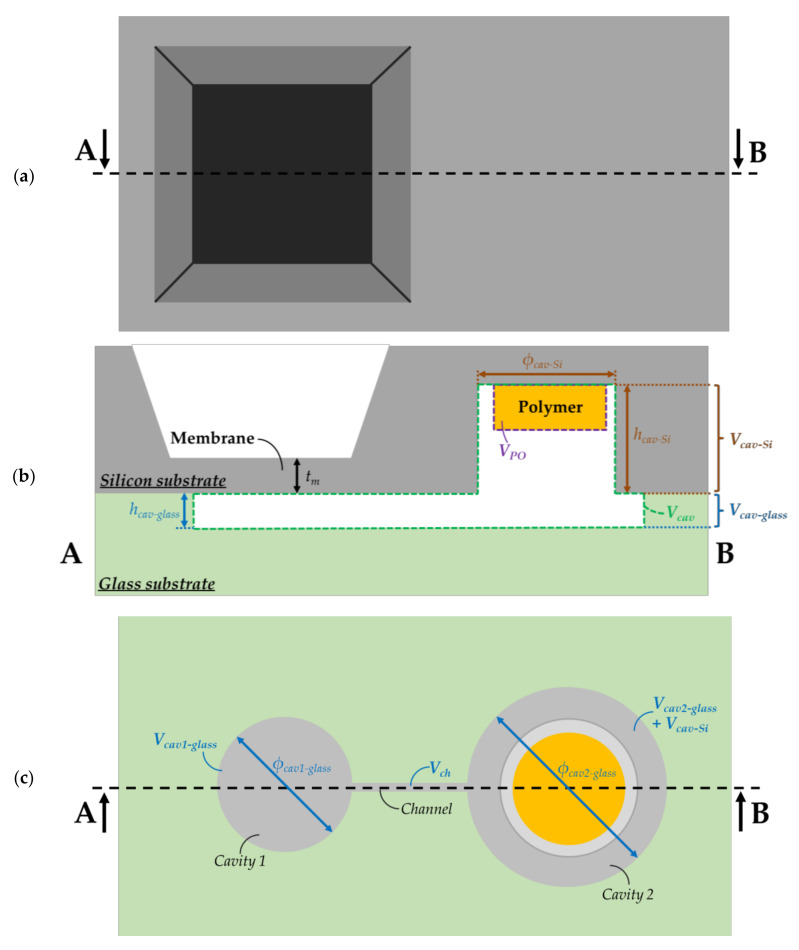
(**a**) Top (i.e., from the silicon structure) (**b**) cross-sectional and (**c**) bottom (i.e., from the glass substrate) view of the investigated microelectromechanical transducer with different volumes. The total sealed structure of volume $V_{cav}$ includes two cavities etched in the silicon and the glass substrates. These cavities are linked by a channel of volume $V_{ch}$. $V_{cav1-glass}$, $h_{cav-glass}$ and $\Phi_{cav1-glass}\;\left(=2r_{m}\right)$ are, respectively, the volume, height and diameter of the circular cavity (*cavity 1*) placed below the membrane of thickness $t_{m}$. The second cavity (*cavity 2*) of volume $V_{cav-PO\;}\left(=V_{cav2-glass}+V_{cav-Si}\right)$ is patterned on the silicon substrate and glass substrate. $\Phi_{cav2-glass}$ denotes the cavity diameter in the glass substrate, while $\Phi_{cav-Si}$ denotes the diameter in the silicon substrate and $h_{cav-Si}$ is the height of the cavity in the Si substrate. $V_{PO}$ designates the polymer volume.

**Figure 4 sensors-21-05912-f004:**
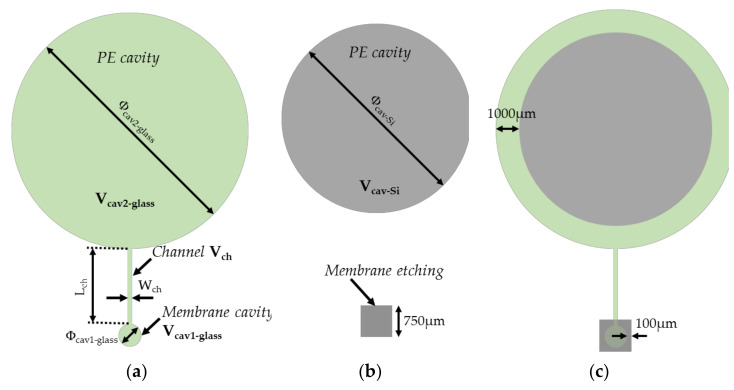
Layouts of the structure on (**a**) the glass substrate and (**b**) the silicon substrate; (**c**) superposition of the two layouts.

**Figure 5 sensors-21-05912-f005:**
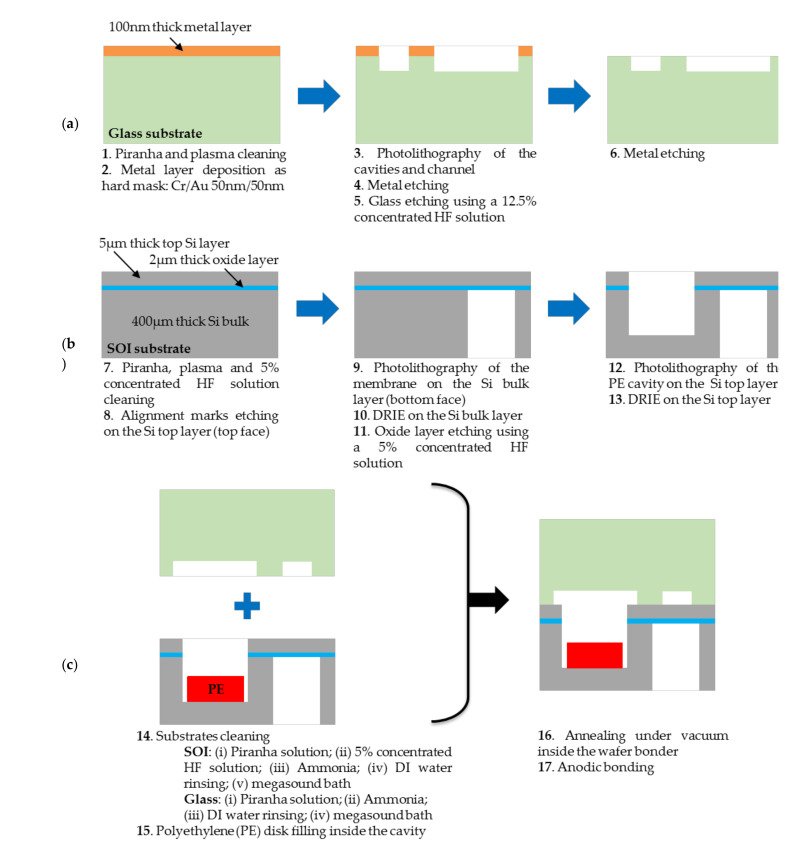
Steps of the fabrication process for manufacturing of (**a**) the glass substrate, (**b**) the silicon substrate and (**c**) final anodic bonding.

**Figure 6 sensors-21-05912-f006:**
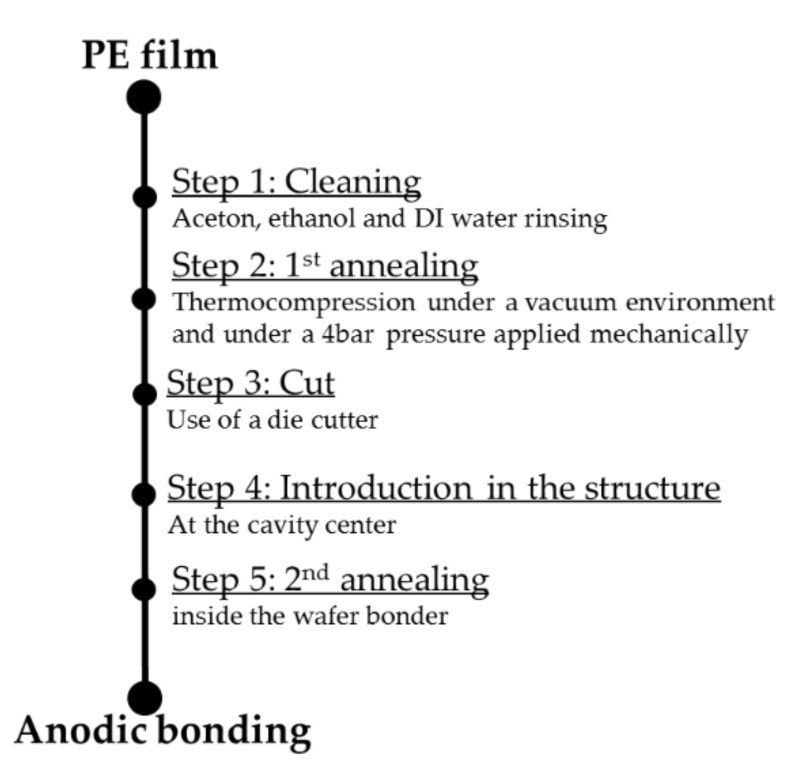
Polyethylene manufacturing process.

**Figure 7 sensors-21-05912-f007:**
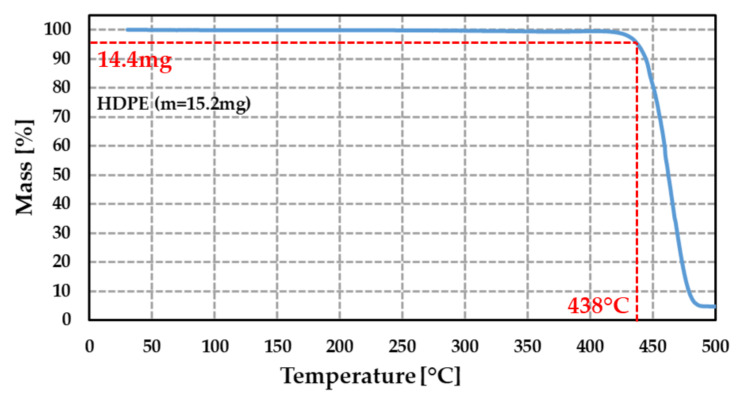
Variation (in %) of the measured mass of HDPE sample as a function of the temperature (initial mass = 15.2 mg). Measurement was performed using a temperature ramp of 5 °C·min^−1^.

**Figure 8 sensors-21-05912-f008:**
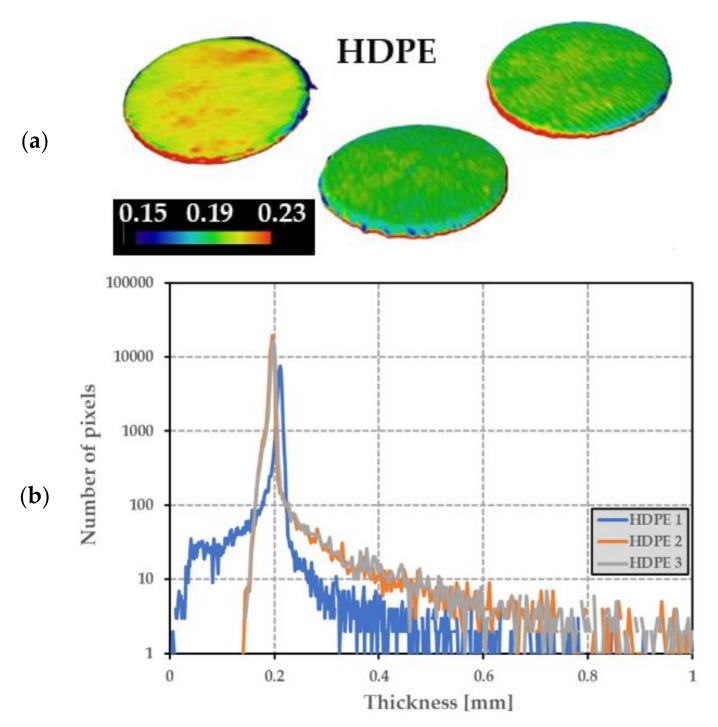
(**a**) Thickness of 3 HDPE disks measured from tomography (color bar is in mm), and (**b**) the number of locations (pixels) on the 3 HDPE disks for which the HDPE thickness takes the values reported in the *x*-axis.

**Figure 9 sensors-21-05912-f009:**
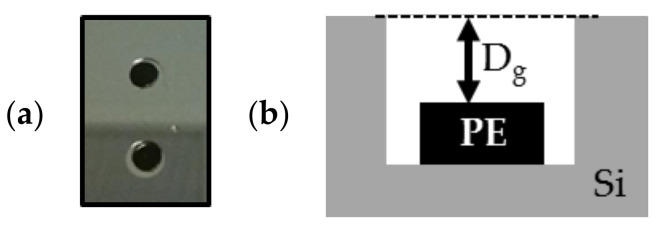
(**a**) Photograph of PE samples (in black) placed in the cavity etched in silicon before annealing; (**b**) schematic of the cross-section of PE sample with the gap length $D_{g}$ between the top of the PE sample and the top of the cavity.

**Figure 10 sensors-21-05912-f010:**
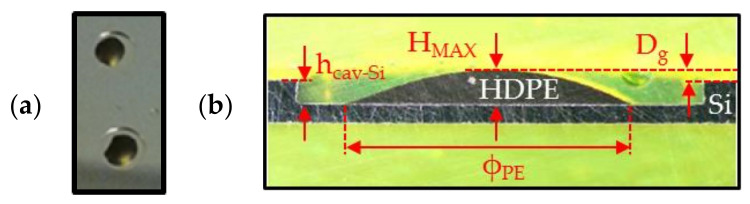
(**a**) Photograph of PE samples (in black) placed in the cavity etched in silicon after annealing; (**b**) microsection photographs of a HDPE sample inside glass and silicon cavities.

**Figure 11 sensors-21-05912-f011:**
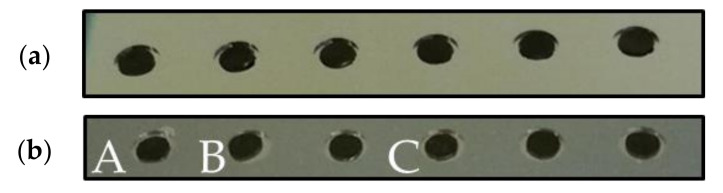
Photographs of the cavities filled with PE samples (**a**) before and (**b**) after annealing and anodic bonding. The microsection of cavities A, B and C are given in Figure 12.

**Figure 12 sensors-21-05912-f012:**
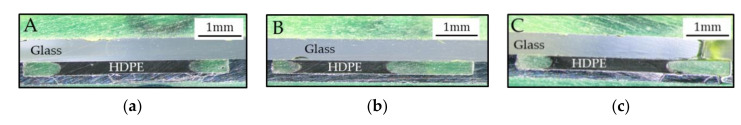
Photographs of the microsection after anodic bonding of the cavities A (**a**), B (**b**) and C (**c**) filled with HDPE samples (see Figure 11).

**Figure 13 sensors-21-05912-f013:**
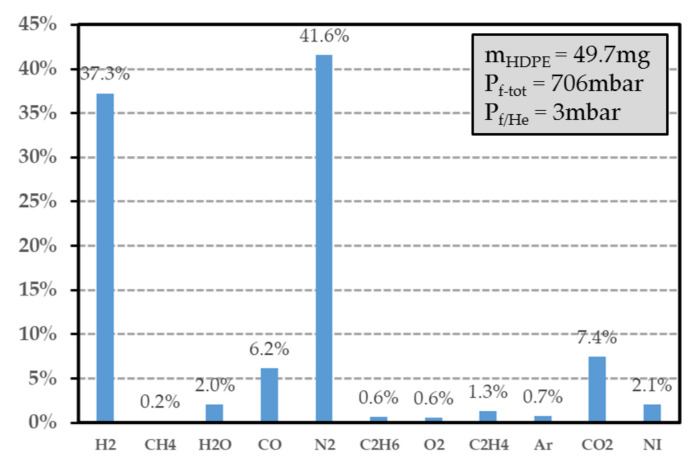
Gas composition of the HDPE sample exposed at a dose level of 24.03 kGy (helium is removed from the spectrogram). The data in the ordinate (in %) represent the volume proportion of each detected gas. NI represents the species that are not identified. $P_{f-tot}$ denotes the pressure after irradiation inside the vial, while $P_{f/He}$ is the final pressure without the helium contribution.

**Figure 14 sensors-21-05912-f014:**
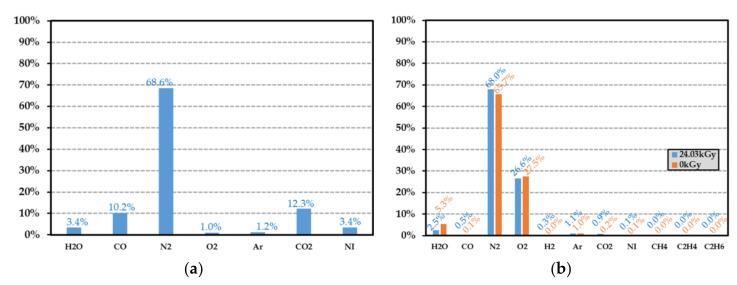
(**a**) Gas composition after a 24.03 kGy dose exposure of a HDPE sample without the hydrogen contribution, and (**b**) gas composition of an empty test vial without polymer before and after an irradiation of 24.03 kGy. The helium contribution is not reported. In subfigure a, the volume proportions of dihydrogen and hydrocarbon molecules are not given. The data in the ordinate (in %) represent the volume proportion of each detected gases.

**Figure 15 sensors-21-05912-f015:**
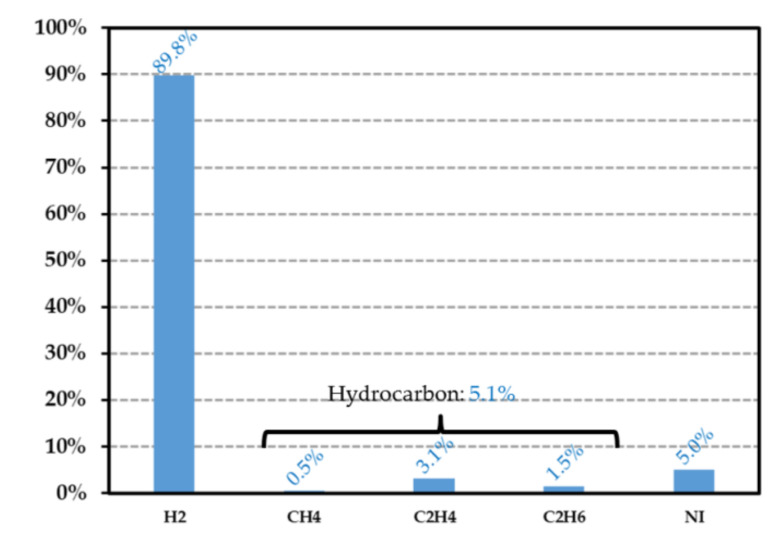
Gas composition after irradiation of a 24.03 kGy HDPE sample. Proportions of He, $O_{2},$ CO, $CO_{2},$ Ar and $N_{2}$ are not reported. The data in the ordinate (in %) represents the volume proportion of each detected gases.

**Figure 16 sensors-21-05912-f016:**
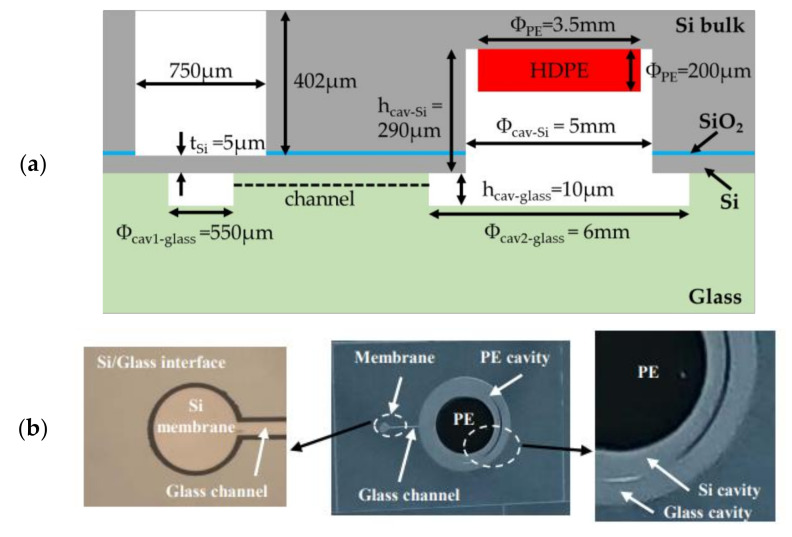
Dimensions of the dosimeter: (**a**) cross sectional view and (**b**) photographs of the cavity fabricated on the glass substrate after anodic bonding [9].

**Figure 17 sensors-21-05912-f017:**
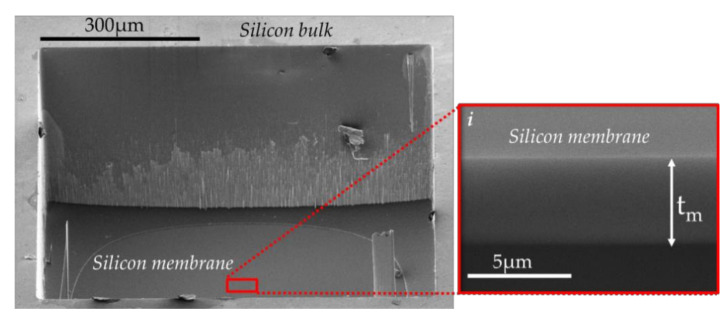
SEM micrograph of the cavity membrane and FIB micrograph showing the membrane cross section (in inset *i*). $t_{m}$ denotes the thickness of the silicon membrane.

**Figure 18 sensors-21-05912-f018:**
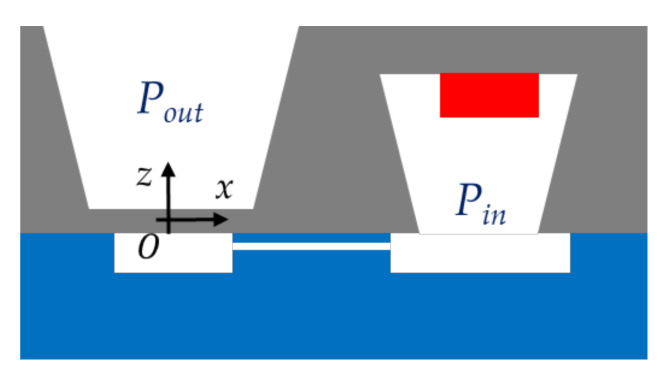
Cross-sectional view of the dosimeter. The deflection amplitude is negative when the pressure $P_{in}$ inside the sealed cavity is lower than the pressure $P_{out}$ outside the cavity.

**Figure 19 sensors-21-05912-f019:**
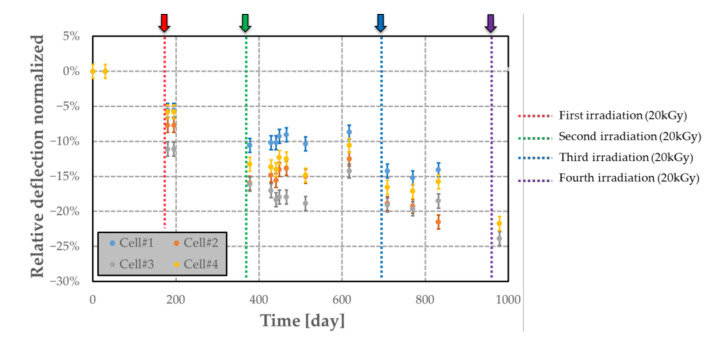
Airtightness variation over time of passive MEMS transducers (with its polymer) subjected to several nuclear radiations. Each arrow indicates when the irradiation occurs (t = 0 refers to the time of the first deflection measurement).

**Figure 20 sensors-21-05912-f020:**
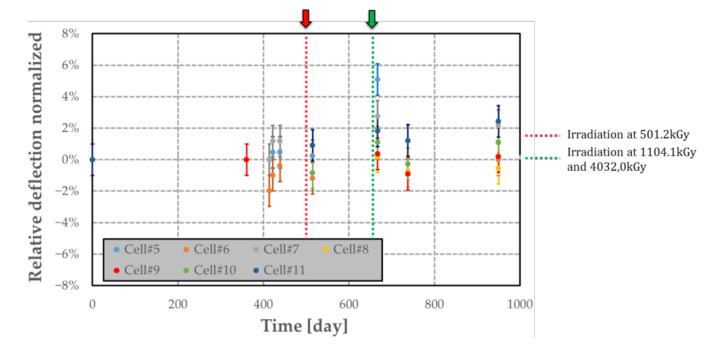
Airtightness variation over time of MEMS transducers (without polymer) subjected to high dose levels. Each arrow indicates when the irradiation occurs (t = 0 refers to the time of the first deflection measurement).

**Figure 21 sensors-21-05912-f021:**
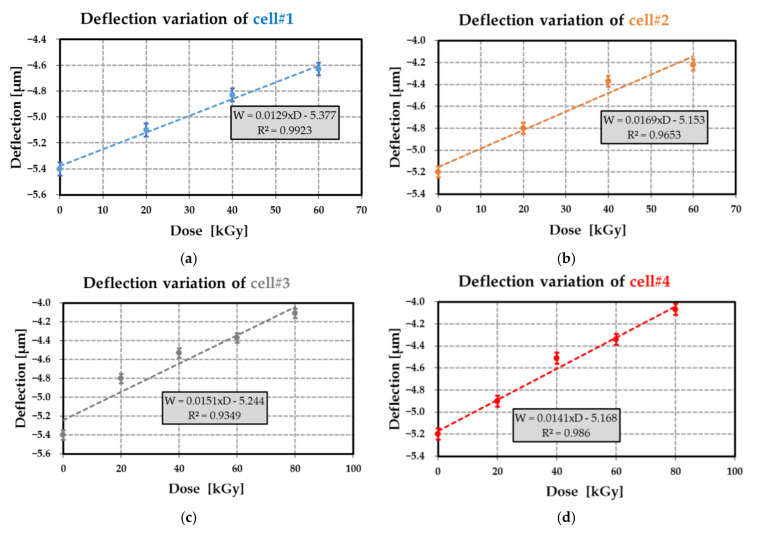
Deflection as a function of the dose level for different passive MEMS test devices ((**a**) for cell#1; (**b**) for cell#2; (**c**) for cell#3 and (**d**) for cell#4). For each transducer the linear regression between the deflection *W* and the dose level *D* are reported (*R*^2^ designates the coefficient of determination of the linear regression).

**Figure 22 sensors-21-05912-f022:**
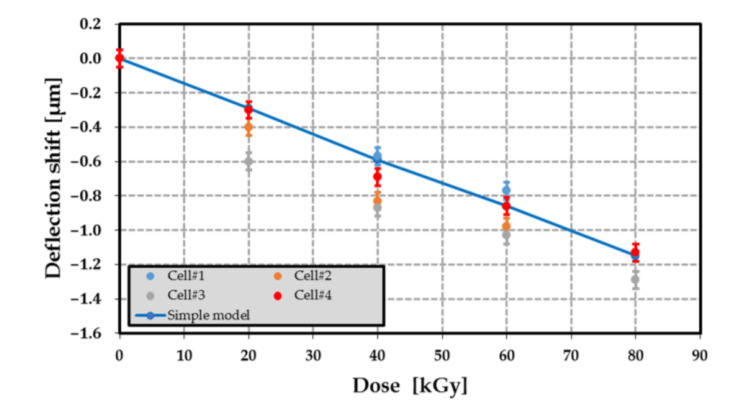
Shift of the deflection as a function of the dose level for different passive MEMS transducers (cell#1 to #4). The blue line refers to the shift computed from the model of Equation (7).

**Figure 23 sensors-21-05912-f023:**
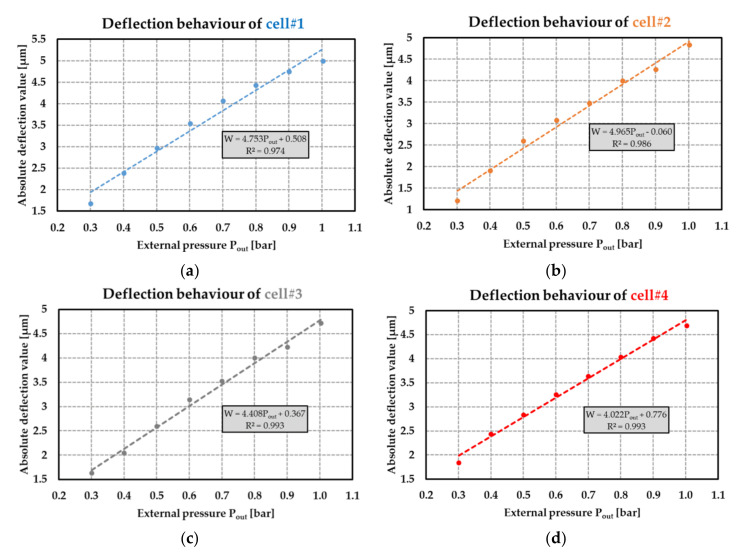
Absolute value of the deflection W with respect to the external pressure $P_{out}$ applied on the MEMS transducers: (**a**) for cell#1; (**b**) for cell#2; (**c**) for cell#3; and (**d**) for cell#4 (*R*^2^ denotes the coefficient of determination of the linear regression).

**Figure 24 sensors-21-05912-f024:**
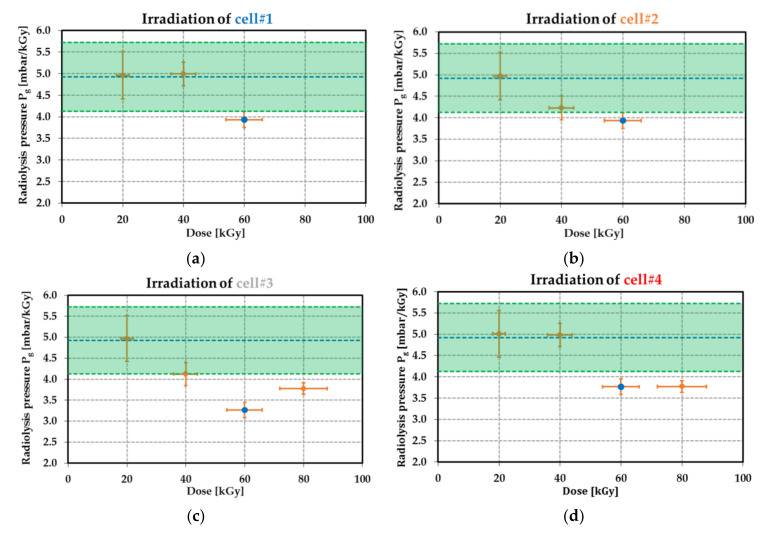
Pressure $P_{g}\;$ generated by the radiolysis as a function of the dose level (in kGy) for each MEMS transducers: (**a**) for cell#1; (**b**) for cell#2; (**c**) for cell#3 and (**d**) for cell#4. The blue point at 60 kGy is estimated from using the results of Figure 23. The horizontal error bars refer to the 10% uncertainty on the dose level (see Table 8), while the vertical error bars correspond to the uncertainty *δP* on the pressure (see (26)). In the green region, the central dashed blue line and dashed green lines correspond respectively to the theoretical value of $P_{g}$ at 4.92 mbar·kGy^−1^ and the uncertainty $\delta P_{g}\;$ of ±0.80 mbar·kGy^−1^ (see Table 17) for the total gas emission yield factor *G* of 4.4 × 10^−7^ mol·J^−1^ (see Table 6).

**Figure 25 sensors-21-05912-f025:**
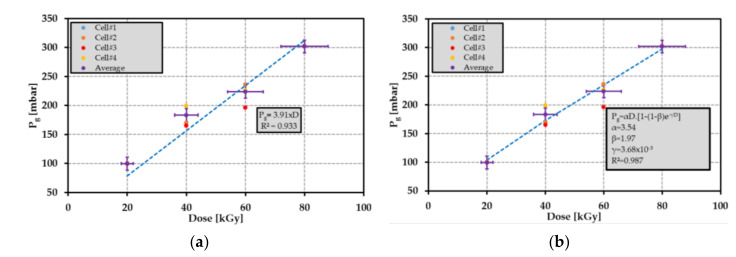
Average pressure $P_{g}$ (in magenta) as a function of dose level using (**a**) a linear relationship and (**b**) the exponential relationship proposed in [95]. The blue, orange, red and yellow points refer to the radiolysis pressure for the cells #1; #2; #3 and #4, respectively.

**Table 1 sensors-21-05912-t001:** Height (in µm), diameter (in µm) and volume (in mm^3^) of cavities patterned on the glass substrate (see Figure 4). *W_ch_* (in µm) and *L_ch_* (in µm) denote, respectively, the width and length of the channel between the two cavities.

*h_cav-glass_*	*ϕ* _*cav*1*-glass*_	*L_ch_*	*W_ch_*	*ϕ* _*cav*2*-glass*_	*V* _*cav*1*-glass*_	*V_ch_*	*V* _*cav*2*-glass*_	*V_cav-glass_*
10	550	2000	50	6000	2.38 × 10^−3^	1 × 10^−3^	2.83 × 10^−1^	2.86 × 10^−1^

**Table 2 sensors-21-05912-t002:** Height (in µm), diameter (in µm) and volume (in mm^3^) of the cavity patterned on the silicon substrate, and the ratio $V_{PO}/V_{g}$ ratios for HDPE samples of 3.5 mm diameter with two different thicknesses $t_{PE}$ of 10 µm and 200 µm.

*ϕ_cav-Si_*	$t_{PE}$	*h_cav-Si_*	*V_PO_*	*V_cav-Si_*	*V_g_*	*V_PO_/V_g_*
5000	10	60	9.62 × 10^−2^	1.18	1.37	7.02%
	200	250	1.92	4.91	3.27	58.7%

**Table 3 sensors-21-05912-t003:** Mean thickness values for three different HDPE disks.

Polymer	Mean Thickness Value (mm)	Mean Total Value (mm)	Standard Deviation (%)
HDPE #1	0.2108	0.201	4.2%
HDPE #2	0.1954		
HDPE #3	0.1973		

**Table 4 sensors-21-05912-t004:** Gap length $D_{g}$ between the top of the PE sample and the top of the cavity, and maximum thickness $H_{MAX}$ of PE sample inside the cavity after the annealing process. $\Delta H_{MAX}$ is the maximum thickness variation of the PE sample before and after annealing, while $\Phi_{PE}$ is the diameter of the PE sample after annealing.

Polymer	*Φ_PE_* (mm)	*h_cav-Si_* (µm)	*H_MAX_* (µm)	*D_g_* (µm)	Δ*H_MAX_* (µm)
HDPE#1	3.5	287	470	183	269
HDPE#2	3.8	293	424	131	223

**Table 5 sensors-21-05912-t005:** Typical gas composition in ambient environment.

N_2_	O_2_	Ar	H_2_O
78.1%	21.0%	0.9%	Between 0.5% and 5% *

* According to the humidity level.

**Table 6 sensors-21-05912-t006:** Emission yield factors (in 10^−7^ mol·J^−1^) of each gas constituent after irradiation of an HDPE sample of 49.7 mg. The factor $G_{gas}$ (in 10^−7^ mol·J^−1^) is the total emission yield factor and was calculated from summation of the emission yield factors of all gas constituents. DL stands for “quantity smaller than the Detection Limit of the apparatus” and NI stands for “Not Identified species”.

PE Material	G_H2_	G_CH4_	G_C2H4_	G_C2H6_	NI	*G_gas_*
HDPE (24.03 kGy irradiation)	4.0	<DL	0.1	0.1	0.2	4.4

**Table 7 sensors-21-05912-t007:** Hydrogen emission yield factor (in 10^−7^ mol·J^−1^) for different HDPE samples irradiated at two different dose levels and for various film thicknesses.

Thickness and Sample Material	12.15 kGy	24.03 kGy
HDPE 10 µm thick	4.1	4.3
HDPE 200 µm thick	3.9	4.0
HDPE 1 mm thick	3.7	4.1

**Table 8 sensors-21-05912-t008:** Measurement uncertainties on the gas emission yield factors.

X	${\%}_{x}^{vol}$	*P_f_* [mbar]	*V_amp_* [cm^3^]	*m_PE_* [mg]	*T* [K]	*D* [kGy]	$G_{x}[\mathrm{mol}\cdot J{-}^{1}]$
ΔX		1	0.27	0.1	1		
ΔX/X	1%	0.14%	2.7%	0.22%	0.33%	10%	10.4%

**Table 9 sensors-21-05912-t009:** Average value and standard deviations of 20 HDPE mass measurement results obtained by using a microbalance and performed on the same sample (reproducibility analysis).

Average Value (mg)	Absolute Standard Deviation (Δm, in mg)	Relative Standard Deviation (Δm/m, in %)
1.77	0.04	2.3

**Table 10 sensors-21-05912-t010:** Average value and standard deviations of 35 mass measurement results obtained by using a microbalance (variability analysis).

Average Value (mg)	Absolute Standard Deviation (Δm, in mg)	Relative Standard Deviation (Δm/m, in %)
1.78	0.09	5.1

**Table 11 sensors-21-05912-t011:** Average and standard deviation of measured height of cavities etched into the glass substrate.

# Substrate	1	2	3	4	Overall Average
*ĥ_cav-glass_* [µm]	8.890	8.952	8.730	8.870	8.859
Δ*h_cav-glass_* [µm]	0.090	0.079	0.097	0.089	0.089
Δ*h_cav-glass_/h_cav-glass_*					1.0%

**Table 12 sensors-21-05912-t012:** Relative and absolute uncertainties on the cavity volume performed on the glass substrate.

*h_cav-glass_* [µm]	$\phi_{cav2}^{over}[µm]$	$\Delta \phi_{cav2}^{over}[µm]$	*V*_*cav*2*-glass*_ [mm^3^]	ΔV_cav2-glass_ [mm^3^]	$\frac{\Delta V_{cav2-glass}}{V_{cav2-glass}}[\%]$
10	6016.21	2.13	2.827 × 10^−1^	2.530 × 10^−3^	0.9

**Table 13 sensors-21-05912-t013:** Average and standard deviation of measured height of cavities etched into the silicon substrate.

# Substrate	1	2	3	4	Overall Average
ĥ_cav-Si_ [µm]	290.50	320.22	310.80	304.60	304.28
Δh_cav-Si_ [µm]	1.29	2.84	2.64	1.34	2.02
Δh_cav-Si_/h_cav-Si_					0.7%

**Table 14 sensors-21-05912-t014:** Relative and absolute uncertainties on the cavity volume performed on the silicon substrate.

*h_cav-Si_* [µm]	$\phi_{cav-Si}^{th}[µm]$	$\Delta \phi_{cav-Si}[µm]$	*V_cav-Si_* [mm^3^]	Δ*V_cav-Si_* [mm^3^]	$\frac{\Delta V_{cav-Si}}{V_{cav-Si}}[\%]$
290	5000	2	5.694	3.986 × 10^−2^	0.7

**Table 15 sensors-21-05912-t015:** Relative and absolute uncertainties on the sealed cavity volume *V_g_*.

*V_g_* [mm^3^]	Δ*V_g_* [mm^3^]	$\frac{\Delta V_{g}}{V_{g}}[\%]$
4.044	0.214	5.3

**Table 16 sensors-21-05912-t016:** Relative uncertainties ΔX/X on the parameter X appearing in (16).

X	$G_{gas}$	*m_PE_*	$V_{g}$	*T*	*D*	$P_{g}$
ΔX/X	10.4%	5.1%	5.3%	0.3%	10.0%	16.2%

**Table 17 sensors-21-05912-t017:** Theoretical determination of the pressure $P_{g}$ generated by polymer radiolysis during nuclear irradiation and membrane deflection $\Delta W_{g}$ for five different dose levels. $\Delta P_{g}$ denotes the absolute uncertainty on pressure $P_{g}$.

Dose Level [kGy]	*P_g_* [mbar]	Δ*P_g_* [mbar]	Δ*W_g_* [µm]
1	4.92	0.80	0.01
20	98.44	15.95	0.29
40	196.88	31.89	0.57
60	295.32	47.84	0.86
80	393.76	63.79	1.15

**Table 18 sensors-21-05912-t018:** Composition, partial pressure (in mbar) inside (noted in.) and outside (noted out.) of the cavity and permeation coefficient (in cm^2^·s^−1^) through silicon ($K_{X}^{Si}$) and glass ($K_{X}^{SiO_{2}}$) for some components X at a given temperature. $\left|\Delta P_{X}\right|$ corresponds to the difference of partial pressure (in mbar) for each gas component inside and outside the cavity.

Gas Component	Gas Composition (out.)	Partial Pressure (out.)	Gas Composition (in.)	Partial Presssure (in.)	$\left|\Delta P_{X}\right|$	$K_{X}^{SiO_{2}}$	$K_{X}^{Si}$
$N_{2}$	78.1%	781	0%	0	781	5.0 × 10^−17^ (25 °C)	<1.9 × 10^−11^ (1200 °C)
$O_{2}$	21.0%	210	0%	0	210	5.0 × 10^−17^ (25 °C)	NR
Ar	0.9%	9	0%	0	9	<10^−15^ (700 °C)	<1.9 × 10^−11^ (1200 °C)
$CO_{2}$	4.2 × 10^−2^%	4.2 × 10^−1^	0%	0%	4.2 × 10^−1^	NR	NR
$H_{2}O$	0.5–5%	5–50	0%	0	5–50	NR	NR
$C_{x}H_{y}$	1.8 × 10^−4^% *	1.8 × 10^−3^ *	5.1% *	5.1 *	5.1 *	NR	NR
$H_{2}$	5.5 × 10^−5^%	5.5 × 10^−4^	89.8%	898.0	898	3.4 × 10^−16^ (20 °C)	5.7 × 10^−40^ (20 °C)
He	5.2 × 10^−4^%	5.2 × 10^−3^	0%	0	5.2 × 10^−3^	3.5 × 10^−11^ (20 °C)	3.2 × 10^−34^ (20 °C)
Ne	1.8 × 10^−3^%	1.8 × 10^−2^	0%	0	1.8 × 10^−2^	2 × 10^−16^ (25 °C)	<1.9 × 10^−11^ (1200 °C)
Kr	1.1 × 10^−4^%	1.1 × 10^−3^	0%	0	1.1 × 10^−3^	NR	NR

* Mainly $CH_{4}\;$; NR stands for not reported in the literature according to the best knowledge of the authors.

**Table 19 sensors-21-05912-t019:** Diffusion coefficient at room temperature noted *D_room_* for some chemical species in amorphous silicon dioxide [102].

Molecule	*D_room_* [cm^2^·s^−1^]
He	2.8 × 10^−8^
$H_{2}$	1.5 × 10^−11^
$H_{2}O$	5.3 × 10^−20^

**Table 20 sensors-21-05912-t020:** Partial pressure difference, permeation rate through the glass material and leak rate for each gas component.

Gas Component	$\left|\Delta P_{X}\right|[\mathrm{mbar}]$	$K_{X}^{SiO_{2}}[{\mathrm{cm}}^{2}\cdot s^{-1}]$	$L_{X}^{SiO_{2}}[\mathrm{bar}\cdot {\mathrm{cm}}^{3}\cdot s^{-1}]$
$N_{2}$	781	5.0 × 10^−17^ (25 °C)	8.8 × 10^−16^
$O_{2}$	210	5.0 × 10^−17^ (25 °C)	2.4 × 10^−16^
$H_{2}$	898	3.4 × 10^−16^ (20 °C)	6.9 × 10^−15^

**Table 21 sensors-21-05912-t021:** Time (in years) necessary to see a difference δP inside the cavity.

Gas Component	δP=50 mbar	δP=10 mbar	δP=5 mbar	δP=1 mbar
$N_{2}$	~7600	~1470	~730	~150
$O_{2}$	~30,620	~5500	~2710	~540
$H_{2}$	~960	~190	~94	~19

## Data Availability

Not applicable.

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
