# Peer review of "Microelectromechanical Transducer to Monitor High-Doses of Nuclear Irradiation"

_sensors, 2021, doi:10.3390/s21175912_

Round 1

Reviewer 1 Report

The authors report a passive microelectromechanical transducer for the wireless monitoring of high irradiation doses in nuclear environments. The sensor performances are validated experimentally up to 80kGy and are in good agreement with theoretical predictions. This manuscript could be published in Sensors.

Author Response

Response to Reviewer 1

Comments

The authors report a passive microelectromechanical transducer for the wireless monitoring of high irradiation doses in nuclear environments. The sensor performances are validated experimentally up to 80kGy and are in good agreement with theoretical predictions. This manuscript could be published in Sensors.

Authors’ reply:

We thank the Reviewer for his/her positive feedback.

Reviewer 2 Report

This paper presents the underlying concept, design and testing of new radiation sensor based on radiation-induced degradation of polymer materials. In essence, the sensor works because the polymer is encased inside of membrane cavity such that the increase in pressure of cavity by degradation of the polymer results in visible deflection. The idea is excellent and the paper presents very detailed description of analysis conducted using prototypes. It seems that the sensor is working convincingly well. One minor suggestion: this type is sensor is the "cumulative dose" sensor because the gas generated by radiation damage stays inside of the membrane. One possible source of error is then the escape of the gases from the cavity. According to the gas analysis, the majority of gases is CHx type so they may not escape well via membrane. However, they will eventually generate H2 that can permeate through the membrane. It may be good to mention such limitations.

Reviewer 3 Report

In this manuscript, the authors presented a design for high-dose nuclear irradiation. The authors utilized the gassing properties of a polymer material upon nuclear irradiation to monitor the radiative dose.  The authors first fabricated a MEMS device and sealed the HDPE polymer material inside a cavity. Upon radiation, the gas released form the polymer was detected by the gas emission factor. The authors also carefully examined the relationship between the gas release rate and radiative doses. Theoretical evaluation was also proceeded to validate the experimental results as well. Although the design was similar to the authors’ previous publications, some new gas releasing data and influences of design factors were provided with more detailed explanation. The manuscript is also written in a good organization, and thus is okay for publication. Some suggestions are listed for the authors to perfect the manuscript:

  1. The RF resonator measurements for deflection is not quite clear. The authors addressed the issues in previous publication, but it is not easy for readers to understand. Please elucidate in line 74-79 and Figure 1 to avoid readers’ confusion.
  2. What is the detection limit or minimum radiative dose for the sensor?
  3. How stable is the device subjected to thermal temperature variation? Although TGA in Figure 7 shows the HDPE has a degradation temperature > 400 degree C, the thermal stability of the whole device should also be evaluated.
  4. A comparison between this and other devices should be provided to address the advantages of this sensing strategy.
